# Analysis of subunit folding contribution of three yeast large ribosomal subunit proteins required for stabilisation and processing of intermediate nuclear rRNA precursors

**Gisela Pöll**[1], **Michael Pilsl**[2], **Joachim Griesenbeck**[1]*, **Herbert Tschochner**[1]*, **Philipp Milkereit**[1]*

**1** Chair of Biochemistry III, Regensburg Center for Biochemistry, University of Regensburg, Regensburg, Germany, **2** Structural Biochemistry Unit, Regensburg Center for Biochemistry, University of Regensburg, Regensburg, Germany

\* joachim.griesenbeck@vkl.uni-regensburg.de (JG); herbert.tschochner@vkl.uni-regensburg.de (HT); philipp.milkereit@vkl.uni-regensburg.de (PM)

## Abstract

In yeast and human cells many of the ribosomal proteins (r-proteins) are required for the stabilisation and productive processing of rRNA precursors. Functional coupling of r-protein assembly with the stabilisation and maturation of subunit precursors potentially promotes the production of ribosomes with defined composition. To further decipher mechanisms of such an intrinsic quality control pathway we analysed here the contribution of three yeast large ribosomal subunit r-proteins rpL2 (uL2), rpL25 (uL23) and rpL34 (eL34) for intermediate nuclear subunit folding steps. Structure models obtained from single particle cryo-electron microscopy analyses provided evidence for specific and hierarchic effects on the stable positioning and remodelling of large ribosomal subunit domains. Based on these structural and previous biochemical data we discuss possible mechanisms of r-protein dependent hierarchic domain arrangement and the resulting impact on the stability of misassembled subunits.

## Introduction

Cellular production of ribosomes involves the synthesis of the ribosomal RNA (rRNA) and protein (r-protein) components and their accurate assembly and folding. The rRNAs are initially synthesized as part of precursor transcripts (pre-rRNA) which are extensively processed and modified. In the yeast *S. cerevisiae* (subsequently called yeast) as in other eukaryotic model organisms most of these steps take place in the nucleus, thus requiring subsequent nucleo-cytoplasmic transport of the largely matured small and large ribosomal subunit (SSU and LSU, respectively) precursors (reviewed in [1]). In yeast, over 200 factors have been identified which promote the timely coordinated progression of the numerous SSU and LSU maturation events. Most of these factors interact transiently at defined maturation states either directly or as component of larger modules with the subunit precursors. This leads to the formation of a series of

**Data Availability Statement:** Structure models are available from the wwPDB database (accession numbers 7OF1, 7OH3, 7OHQ, 7OHR, 7OHS, 7OHT,

7OHU, 7OHV, 7OHP, 7OHW, 7OHX and 7OHY). EM density maps are available from the EMDB database (accession numbers EMD-12866, EMD-12892, EMD-12905, EMD-12906, EMD-12907, EMD-12908, EMD-12909, EMD-12910, EMD-12904, EMD-12911, EMD-12912 and EMD-12913). Original unprocessed EM micrographas are available from the EMPIAR database (accession numbers are EMPIAR-10776, EMPIAR-10780, EMPIAR-10774 and EMPIAR-10775).

**Funding:** This work was supported by a grant which was given in the collaborative research center SFB 960, from the "Deutsche Forschungsgemeinschaft" (www.dfg.de) to JG, HT and PM. The cryo-EM-facility of the Julius-Maximilian University Würzburg was supported by the "Deutsche Forschungsgemeinschaft" (INST 92/903-1FUGG). UCSF ChimeraX, which was used in this work, is developed by the Resource for Biocomputing, Visualization, and Informatics at the University of California, San Francisco, with support from National Institutes of Health R01-GM129325 and the Office of Cyber Infrastructure and Computational Biology, National Institute of Allergy and Infectious Diseases. The funders had no role in study design, data collection and analysis, decision to publish, or preparation of the manuscript.

**Competing interests:** The authors have declared that no competing interests exist.

intermediary subunit precursor particles whose composition, rRNA processing states and three-dimensional structures have been deduced in the past (reviewed in [2–6]).

The structure determination of a wide range of nuclear LSU precursor populations isolated from yeast wildtype or mutant strains indicated that the orientations of the seven major LSU rRNA secondary structure domains [7] (subsequently called LSU rRNA domains) with respect to each other are stabilized in a specific order [8–13]. In these LSU precursors the folding states within stably oriented domains, including the associated r-proteins, appear already to a large extent as observed in mature ribosomes. Structure probing experiments indicated that LSU precursors adopt prevalently flexible conformations during the very initial stages [14] with some of the respective RNA-RNA and RNA-protein interactions being progressively established [15]. Downstream of this, the earliest particles amenable to comprehensive tertiary structure analyses by single particle cryo-electron microscopy (cryo-EM) showed a stably folded core of LSU rRNA domains I and II [9]. A defined folding state and orientation of LSU rRNA domains VI and III towards the early core particle was observed in early intermediate nuclear pre-LSU populations [9,10,13]. Late intermediate nuclear particles were characterized by progressive positioning of LSU rRNA domains IV and V and of the 5S rRNA in context of the previous domains [11,16]. Downstream of this, at late nuclear maturation stages, the subunit fold is remodelled by a near 180˚C rotation of the 5S RNP into its mature orientation [8]. Final maturation, including further stabilisation and refolding of the peptidyl transferase centre at the LSU subunit interface is thought to continue after nuclear export in the cytoplasm [12,17–19]. All these steps are accompanied by the timely coordinated association and removal of factors.

The deduced sequence of the numerous LSU folding, assembly, precursor rRNA (pre-rRNA) processing and factor recruitment and release events is thought to be largely caused by hierarchical interrelationships between them. In support of this, the efficient nuclear export and the progression of many of the pre-rRNA processing steps depend on the ongoing r-protein assembly in specific ribosomal subregions [20]. In case of the yeast LSU three groups of r-proteins could be distinguished whose availability is required for either early (~ phenotype group 1), intermediate (~phenotype group 2) or late nuclear pre-rRNA processing steps (~phenotype group 3) [21]. Ongoing expression of a fourth group of r-proteins is largely dispensable for these pre-rRNA maturation steps but still affects, as the others, the stability and nuclear export of LSU precursors. For each of these r-protein groups the members bind to up to a few specific subregions of the LSU, often with direct interactions in between the respective r-proteins (see Figures A and B in S1 Appendix for an overview). The functional role of r-protein of phenotype groups 1–3 highly correlates with their step wise appearance within respective rRNA domains as structurally resolvable regions in nuclear LSU precursor populations. Group 1 proteins are resolvable together with LSU rRNA domains I and II and the core of domain VI in early LSU precursor structures. Group 2 proteins are seen at latest together with LSU rRNA domain III and the rest of LSU rRNA domain VI in density maps of early intermediate LSU precursors. And group 3 r-proteins can be traced in maps of late intermediate nuclear LSU precursors together with the 5S RNP and LSU rRNA domains IV and V. Blocking the expression of individual r-proteins leads to phenotype group specific cooperative assembly effects and changes in the association of ribosome biogenesis factors with immature LSU particles [22–27]. In the examined cases, these effects are evident for, but not restricted to direct protein interaction partners of the *in vivo* depleted r-proteins.

Functional coupling of r-protein assembly with the stabilisation and maturation of LSU precursors potentially serves to promote the production of ribosomes with defined composition. To further decipher mechanisms of such an intrinsic quality control pathway we analysed in the present work the impact of selected yeast LSU r-proteins on the progression of LSU

precursor folding. Typically, r-proteins link several rRNA secondary structure elements in one or more rRNA secondary structure domains by direct physical contacts. These interactions have the potential to favour the establishment and stabilisation of the mature tertiary fold in individual RNA secondary structure domains [28] and to orient secondary structure domains towards each other. Such impact on the global ribosomal fold might be further enhanced by an extensive interaction network between r-proteins which is especially evident in the eukaryotic LSU, and interconnects there all the LSU domains (see Figure A in S1 Appendix for an overview) [29]. In the yeast LSU, single domain binders with little protein–protein interactions are enriched among the non-essential r-proteins (see Figure A in S1 Appendix). Consistently, only minor effects on the general rRNA fold were observed by RNA structure probing in yeast LSU's devoid of the low connectivity non-essential single domain binder rpL26 (uL24 according to the r-protein nomenclature published in [30]) [31,32]. By contrast, previous cryo-EM studies revealed significant impact on the SSU and LSU fold by truncations of the essential r-proteins rpS20 (uS10) and rpL4 (uL4), respectively [33,34].

Three yeast r-proteins rpL2 (uL2), rpL25 (uL23) and rpL34 (eL34) were included in the present study which are all required for the stabilisation and processing of intermediary nuclear LSU precursors [21,24,25]. Interestingly, they establish connections within and between rRNA domains each at a clearly different degree (see S1 Movie for an overview): RpL34 primarily binds to LSU rRNA secondary structure domain III and is additionally embedded by protein-protein interactions into an r-protein cluster at domain III (see Figures A and B in S1 Appendix). RpL25 connects as two-domain binder LSU rRNA domains I and III by direct protein-rRNA and by protein-protein contacts (see Figures A and B in S1 Appendix). Both rpL25 and rpL34 are required for intermediate nuclear pre-rRNA processing steps (~ phenotype group 2), namely the initial cleavage in the internal spacer 2 (ITS2) RNA separating 5.8S and 25S rRNA [21,35]. In addition, they were shown to affect the pre-LSU association of rpL2 and rpL43 (eL43) and some factors starting to be seen in structures of late intermediate LSU precursor populations [24,25]. The third selected protein, rpL2 establishes together with rpL43 a protein cluster at the subunit interface with extensive direct contacts to multiple domains (see Figures A and B in S1 Appendix). These are primarily the LSU rRNA domain IV, and in addition domains II, III and V. In structural studies rpL2 and rpL43 could only be visualised starting from late intermediate nuclear stages together with rRNA domain IV. Coincidently, biochemical analyses indicated that their association with LSU precursors is specifically stabilized at intermediate LSU maturation stages concomitant with the initial cleavage in the ITS2 spacer [25]. Both, rpL2 and rpL43 are required for efficient trimming of the ITS2 (~ phenotype group 3, late nuclear LSU rRNA processing) downstream of the initial ITS2 cleavage [21].

We wondered if the three r-proteins rpL2, rpL25 and rpL34 with their different degrees of connectivity in LSUs have a global, a local, or any impact on the yeast pre-LSU folding pathway. And, if these effects might help to further explain their previously observed roles for LSU maturation and stability. To this end, LSU precursor particles were purified from yeast strains in which expression of one of the three r-proteins was shut down. In each case the structures of several particle populations could be resolved by single particle cryo-EM. Based on derived structure models we discuss the possible causes and the functional significance of the observed effects on the yeast LSU pre-rRNA folding pathway.

## Results

To analyse changes in the folding states of intermediate nuclear LSU precursors upon lack of the r-proteins rpL2, rpL25 or rpL34, yeast strains were used which conditionally express one of

these r-proteins under control of the GAL1/10 promoter [24,25]. In addition, in these three strains and in a respective control strain the LSU biogenesis factor Nog1 is chromosomally encoded in fusion with a tandem affinity purification (TAP) tag (see Materials and Methods) [36]. Nog1 is part of a range of intermediate nuclear to early cytoplasmic LSU precursor populations and its association with LSU precursors was previously found to be independent of the ongoing expression of rpL2, rpL25 or rpL34 [25,37].

As expected, shutting down the GAL1/10 promoter in glucose containing medium prevented growth of the three conditional r-protein expression mutants, while all four strains could be cultivated in galactose containing medium (S2 Appendix) [21,38]. Consistent with previous studies, some growth delay was observed in galactose containing medium in the strains expressing the Nog1-TAP fusion protein (S2 Appendix, compare the colony size of the untagged with the one of tagged strains in YPG) [34,39]. LSU particles associated with TAP tagged Nog1 were affinity purified after four hours shift to glucose containing medium. As expected, northern blotting experiments indicated that these contained in the control strain LSU pre-rRNA before (27SA+B pre-rRNA), and after the ITS2 was cleaved (25.5S + 7S pre-rRNA) and removed (25S and 5.8S rRNA) (lanes 2 in Fig 1A–1C) [25,36]. A similar LSU pre-RNA processing state was detected in particles purified after *in vivo* depletion of rpL21 (lanes 6 in Fig 1A–1C). The latter is required for nuclear export but has no obvious role in LSU rRNA processing (~ phenotype group 4) [21,25]. Furthermore, the RNA analyses corroborated previous results [21,24,25] indicating that two consecutive intermediate nuclear LSU pre-rRNA processing steps are inhibited after expression shut down of either rpL25 and rpL34, or of rpL2: initial cleavage in the ITS2 (~ phenotype group 2), and trimming of the ITS2 towards the 5.8S rRNA 3' end (~ phenotype group 3). For rpL25 and rpL34 the ratio of 27S rRNA to its cleavage products (7S and 25.5+25S) was increased in the purified particles (lanes 3–4 in Fig 1A–1C, compare with control strain lanes 2, see also quantitation shown in S8 Appendix) and for rpL2 the ratio of 7S rRNA to the trimmed 5.8S rRNA (lanes 1 and 5 in Fig 1A–1C, compare with control strain lanes 2, see also quantitation shown in S8 Appendix).

## Key folding events of intermediate nuclear LSU precursors can be detected in Nog1-TAP associated particle populations

Single particle cryo-EM analysis followed by particle sorting and three-dimensional structure reconstruction revealed six major populations of LSU precursors in the control sample. These are named in the following Nog1TAP-A, -B, -C, D, -E and Nog1TAP-F (see Material and Methods and S3 Appendix for data acquisition strategies, see S5 Appendix for particle sorting and processing strategies). The obtained maps reached sub-nanometre resolution-range (see S3 Appendix for resolution estimates) allowing for model generation by flexible fitting of previously published nuclear LSU precursor models (see Materials and Methods).

Five of the observed major folding states largely resembled the ones of several previously described nuclear LSU precursor populations. Thus, key events in the intermediate nuclear pre-LSU folding pathway could be recapitulated in structure models derived from the obtained electron density maps (see S2 Movie for an overview of the observed changes in the rRNA fold): In state Nog1TAP-F (see Figs 2A and 3A, see also Figure G in S1 Appendix visually summarizing detected rRNA helices and proteins with their predicted interactions) LSU rRNA domains I, II and VI were observed in a folding state which resembled in large parts the one found in mature ribosomes which are visualized for comparison in Figs 2F and 3F. Additional densities could be attributed to the proximal parts of the ITS2 spacer, and to more than 15 ribosome biogenesis factors and around 20 r-proteins most of which being required for early and intermediate nuclear LSU pre-rRNA processing steps. State Nog1TAP-D (S5 Appendix),

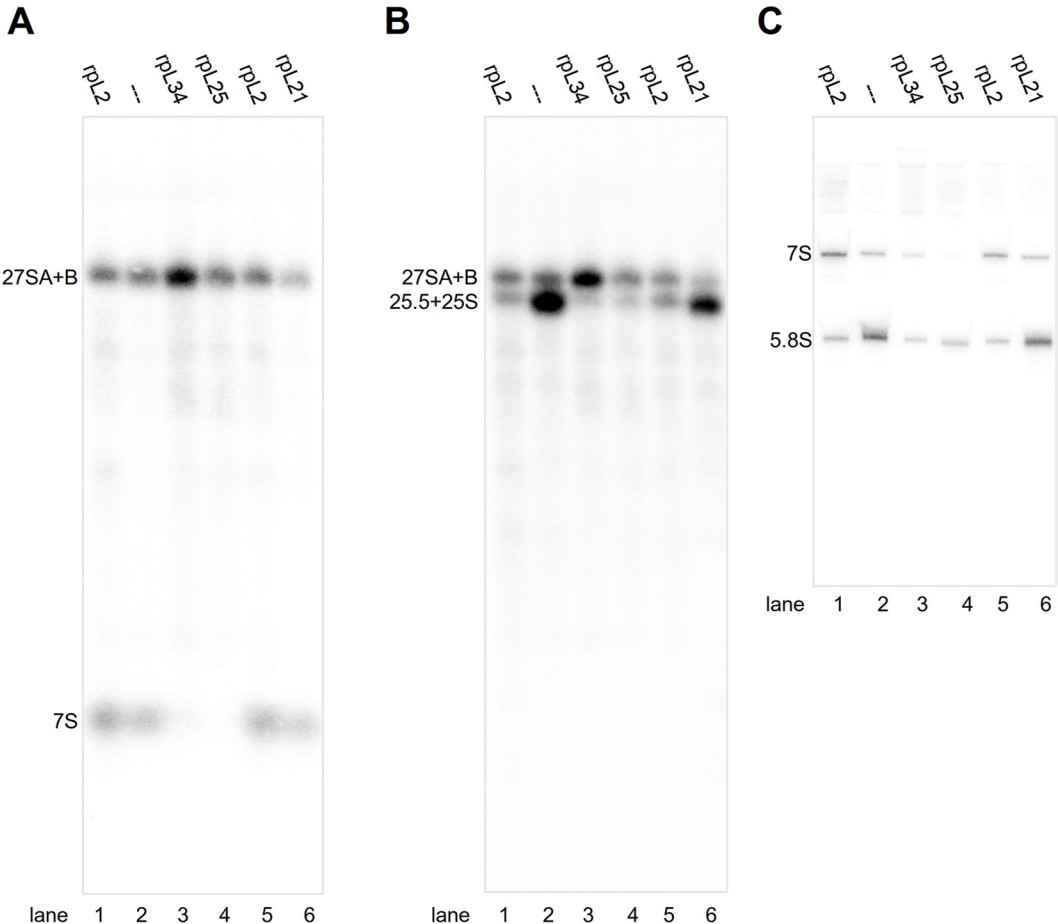

**Fig 1. (Pre-)rRNA composition of Nog1-TAP associated particles purified from yeast conditional r-protein expression mutants.** Cells of yeast strains in which expression of rpL2 (Y1921), rpL21 (Y1813), rpL25 (Y1816), rpL34 (Y2907) or no r-protein (Y1877, label: "—") was shut down for four hours were used as starting material for affinity purification of TAP tagged Nog1. (Pre-)rRNA composition of final eluates was analysed by northern blotting with probes O210 (A), O212 (B) and O209 (C) which detect the (pre-)rRNAs indicated on the left. RNA in (A) and (B) was separated by electrophoresis using an agarose gel and in (C) an acrylamide gel.

for which no molecular model was created resembled state Nog1TAP-F. Still, unlike Nog1-TAP-F it lacked clear densities for the ITS2 spacer RNA and associated factors.

In state Nog1TAP-E (see Figs 2B and 3B and Figure F in S1 Appendix) a group of four factors (Rpf1, Nsa1, Rrp1, Mak16) was possibly released from the subunit. In contrast to state Nog1TAP-F, these could not anymore be detected bound to the rRNA expansion segment ES7 and LSU rRNA domains I and II at the subunit solvent surface (see Figure A in S9 Appendix). Otherwise, a hallmark of state Nog1TAP-E was the appearance of density unambiguously attributable to LSU rRNA domain III and many of the r-proteins associated with it. Among them were rpL25 and rpL34, and other LSU rRNA domain III binders required for the initial cleavage in the ITS2 spacer of LSU pre-rRNA (see Figure B in S9 Appendix). The two-domain binder rpL25 contributes at this stage to the newly formed interaction interface linking LSU rRNA domains I and III.

Starting with state Nog1TAP-C (see Figs 2C and 3C and Figure E in S1 Appendix) densities for large parts of LSU rRNA domains IV and V and the 5S rRNA were detected. Consistent with previous studies the latter was not positioned in this state as in mature ribosomes. It was

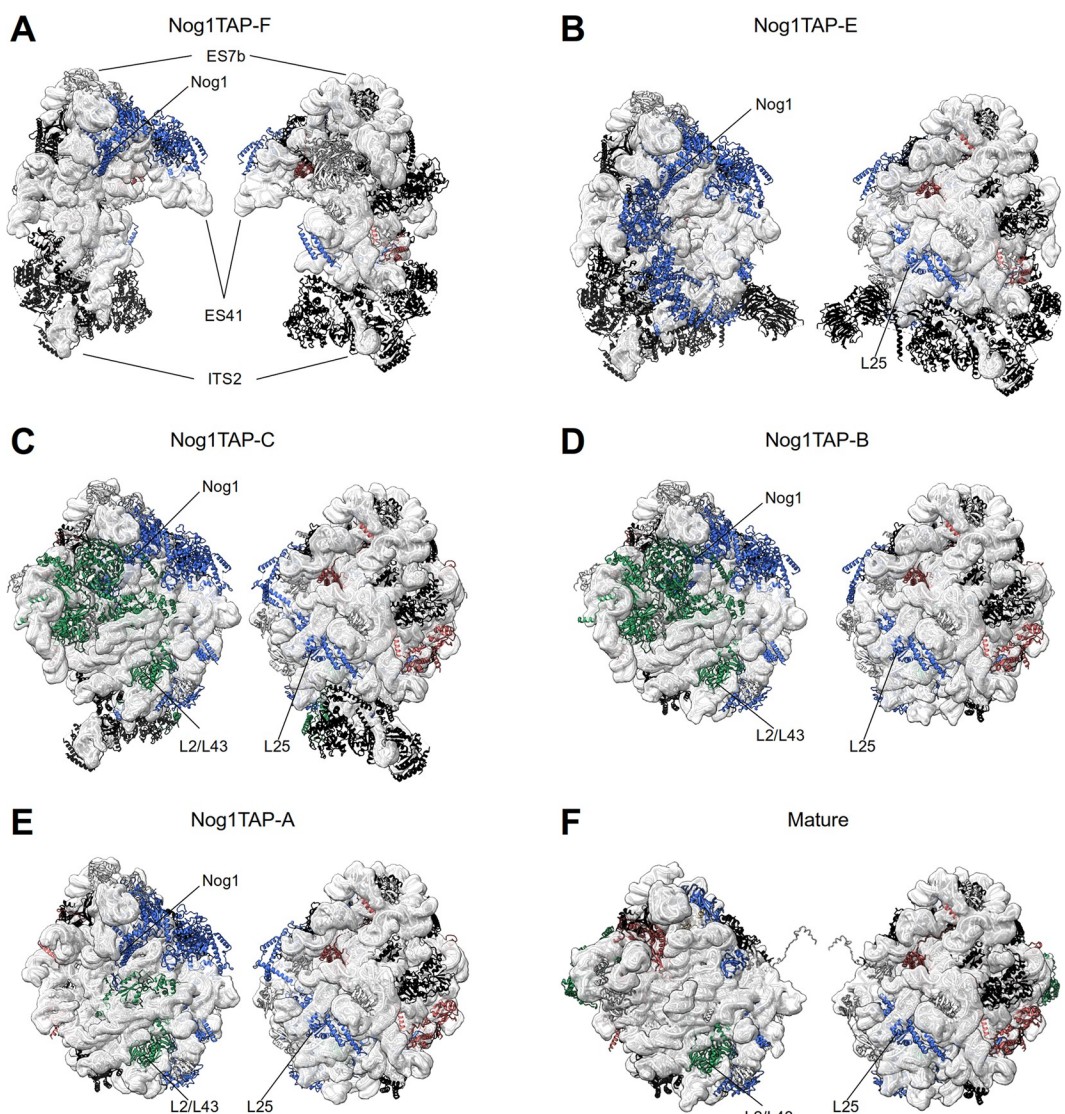

**Fig 2. Structure models of Nog1-TAP associated particle populations from yeast cells with endogenous r-protein expression levels (Y1877).** rRNA of the respective structure models is shown in white cartoon backbone representation and is further highlighted by a transparent model-volume representation (8Å resolution), proteins are shown in cartoon backbone representation with colouring according to functional categories: Proteins required for early pre-rRNA processing are shown in black, for intermediate nuclear pre-rRNA processing steps in blue, for late nuclear pre-rRNA processing steps in green and for downstream nuclear maturation and export in red. Non-essential proteins and proteins with unclear function are shown in grey (see also S6 Appendix). The LSU subunit interface side is shown on the left and the subunit solvent surface side on the right for Nog1TAP-F in (A), for Nog1TAP-E in (B), for Nog1TAP-C in (C), for Nog1TAP-B in (D) and for Nog1TAP-A in (E). Mature large ribosomal subunits are represented in (F). Positions of RNA helices ES7b, ES41 and the ITS2 are indicated in (A) for orientation purposes as well as the positions of Nog1, the rpL2/rpL43 heterodimer and rpL25 if present in the respective model. Detailed protein compositions and model-based protein-protein and protein-RNA interaction-networks are schematically represented in figures A—G in S1 Appendix.

rather rotated by about 180° around its vertical axis. Several factors (Ytm1, Erb1, Noc3, Ebp2, Brx1, Spb1, Nop16) previously associated with LSU rRNA domains I, II and III were not anymore observed and thus possibly released or with flexible orientation (see Figure C in S9 Appendix). Various other factors (Rrs1, Rpf2, Rsa4, Nog2, Nop53, Cgr1) and r-proteins (rpL2, rpL43, rpL5, rpL11, rpL21) newly appeared in Nog1TAP-C in association with the emerging

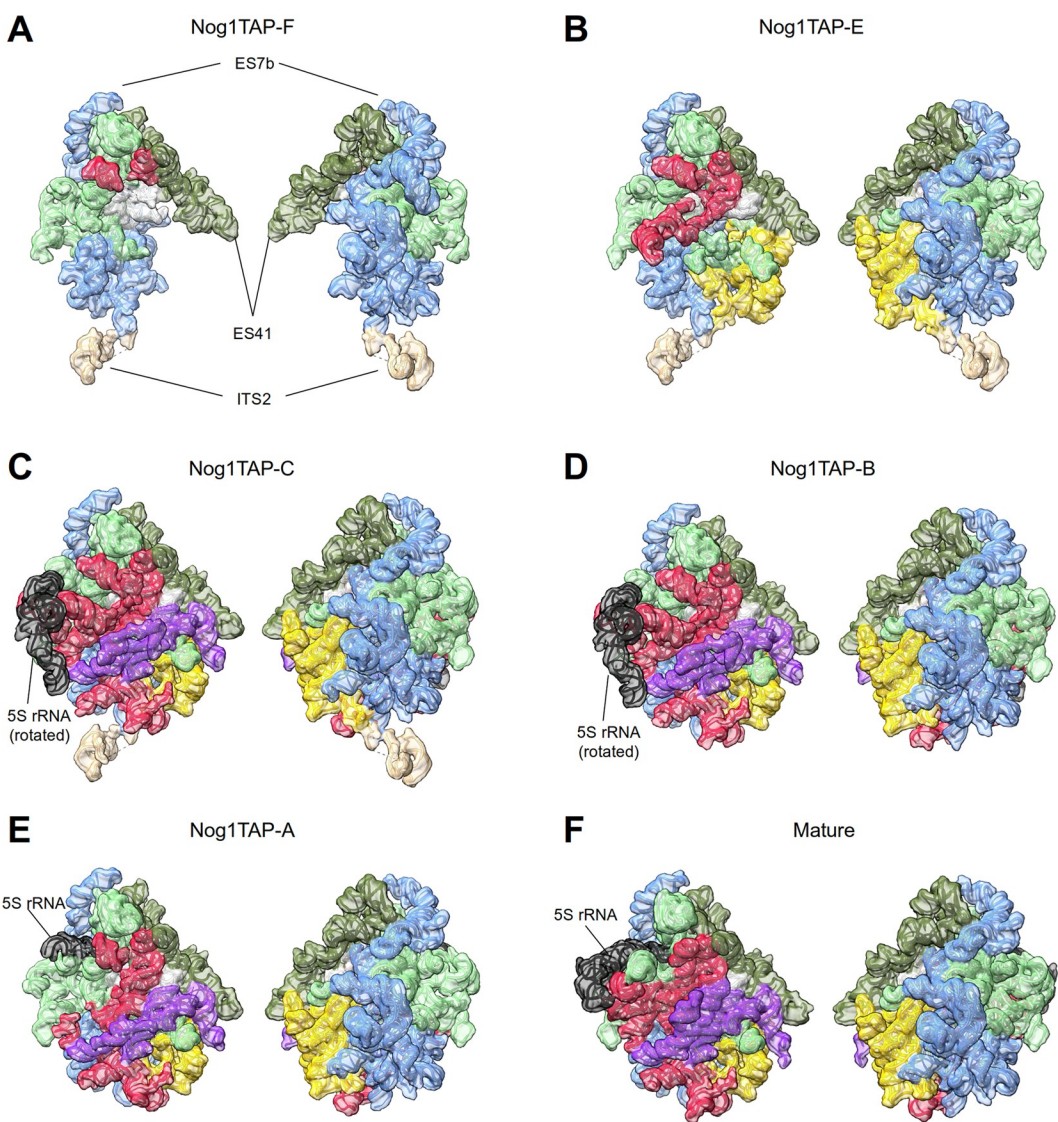

**Fig 3. Folding states of rRNA domains in Nog1-TAP associated particle populations from yeast cells with endogenous r-protein expression levels (Y1877).** rRNA of the respective cryo-EM derived structure models is shown in cartoon backbone representation together with a transparent model-volume representation (8Å resolution). LSU rRNA domain I with expansion segment ES7 is coloured in light blue, domain II in light green, domain III in yellow, domain IV in violet, domain V in red, domain VI in dark green, the 5S rRNA in black and domain 0 in white. The LSU subunit interface side is shown on the left and the subunit solvent surface side on the right for Nog1TAP-F in (A), for Nog1TAP-E in (B), for Nog1TAP-C in (C), for Nog1TAP-B in (D) and for Nog1TAP-A in (E). Mature large ribosomal subunits are represented in (F). Positions of RNA helices ES7b, ES41 and the ITS2 are indicated in (A) for orientation purposes as well as the position of the 5S rRNA if present in the respective model. Presence of individual LSU rRNA helices in the respective structure models is schematically represented in bar diagrams in figures A—G in S1 Appendix.

rRNA domains (see Figure D in S9 Appendix). The multi-domain binder rpL2, together with its binding partner rpL43 is now one of the proteins connecting the previously visible LSU rRNA domains II and III with the newly appearing rRNA of domains IV and V at the subunit interface. Altogether states Nog1TAP-F, Nog1TAP-E and Nog1TAP-C recapitulate previously identified hallmarks of the transition from early intermediate to late intermediate LSU precursors [9].

Starting with state Nog1TAP-B (see Figs 2D and 3D and Figure D in S1 Appendix) no density could be detected anymore for the ITS2 spacer nor for a group of factors associated with it and with LSU rRNA domains I and III (Rlp7, Nop15, Cic1, Nop7, Nop53) (see Figure E in S9 Appendix). Here, trimming of the ITS2 spacer might have led to the release of these factors [40].

Finally, for state Nog1TAP-A (see Figs 2E and 3E and Figure C in S1 Appendix) the density map data support that rotation of the 5S RNP towards its mature position has partially occurred. Densities attributable to 5S rRNA helices 4 and 5 in the mature position were detected and clear densities for factors binding (via rpL5) to the 5S RNP in its premature position were lost (Rsa4, Rrs1, Rpf2) (see Figure F in S9 Appendix).

Apparently, other late nuclear and early cytoplasmic LSU populations with the 5S RNP rotated into its mature position were underrepresented, when considering that untagged Nog1 was previously detected in such particles which associated with carboxy terminal truncated TAP tagged Rlp24 [12]. Carboxy terminal TAP tagging of Nog1 likely interferes with the previously described insertion of Nog1's carboxy terminal domain into the LSU peptide exit tunnel [11,12]. Indeed, corresponding densities could not be detected in Nog1-TAP associated LSU populations. The observed effects on growth rate by TAP tagging of Nog1 (see above, S2 Appendix) and previous LSU folding analyses of particles from Nog1 mutant strains [34] thus suggest that progression of downstream nuclear LSU maturation steps was delayed in this situation to some extent.

## Evidence for global perturbation of the intermediate nuclear folding pathway of LSU particles lacking domain III binders rpL34 or rpL25

This overall representation of intermediate nuclear LSU precursor folding states was not seen for Nog1-TAP associated particles purified from rpL34 expression mutants. Here, only two major folding states (Nog1TAP_L34-A, Nog1TAP_L34-B) were observed in single particle cryo-EM analyses (see Fig 4 and Figures M and N in S1 Appendix, see S3 Movie for an animated overview). For both, the pre-rRNA folding status resembled the one of state Nog1-TAP-F which was observed only for a minor subpopulation of ribosomal particles in the control sample (see processing schemes in S5 Appendix). Defined densities of rRNA domains III, IV and V and the 5S RNA were virtually absent as well as associated factors and r-proteins, including rpL34 itself. In contrast to state Nog1TAP_L34A, in state Nog1TAP_L34B many of the factors were lacking which were bound in Nog1TAP-F to the ITS2 RNA and to the LSU rRNA domains I, II and the ES7 element (compare Figures G, M and N and in S1 Appendix). Similarly, here the densities attributable to parts of the ITS2 spacer RNA could not be detected.

Expression shut down of LSU rRNA domain I and III binder rpL25 led to similar effects on the pre-LSU folding pathway. Again, two major folding states, Nog1TAP_L25-A and Nog1-TAP_L25-B, were detected in Nog1-TAP samples purified from cells where expression of rpL25 was shut down (see Fig 5 and Figures K and L in S1 Appendix, see S4 Movie for an animated overview). Both were, again, characterized by the absence of clear densities for LSU rRNA domains III, IV and V and the 5S rRNA and their associated proteins including rpL25. Hence, their overall fold again closely resembled the one of the low-populated Nog1TAP-F state of the control sample. Only minor displacements of rpL25 main contact sites in LSU rRNA domain I (helices 9 and 10) were observed when compared with states of the control sample showing clear rpL25 densities (see Figure G in S9 Appendix). The factor and r-protein composition of state Nog1TAP_L25-B largely resembled the one observed in state Nog1-TAP-F. On the other hand, in state Nog1TAP_L25-A several factors binding to LSU rRNA domains I and II and the ES7 element could not be detected (Ebp2, Brx1, Rpf1, Nsa1, Rrp1, Mak16) (compare Figures K and L in S1 Appendix).

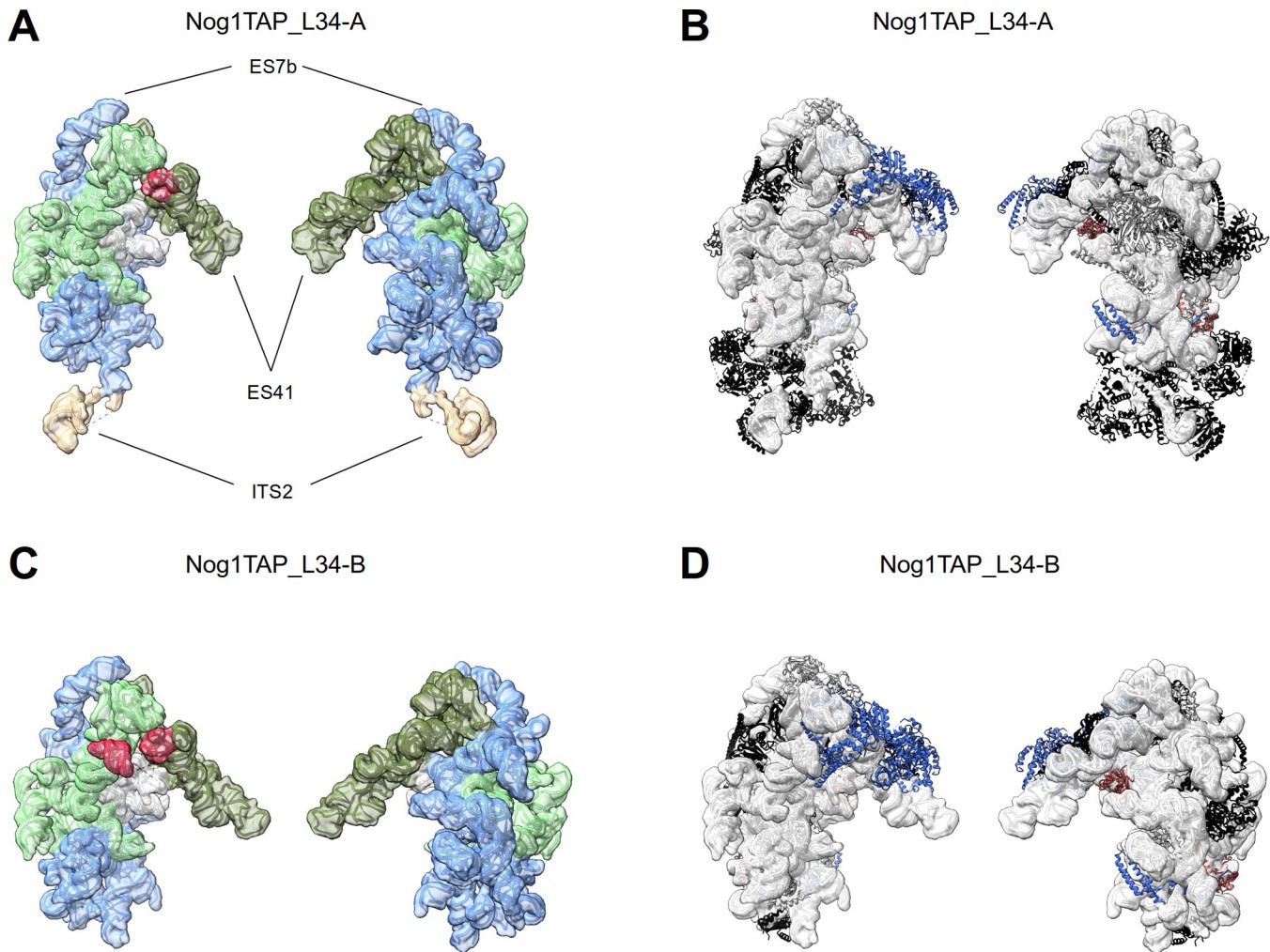

**Fig 4. rRNA folding states and structure models of Nog1-TAP associated particle populations from yeast cells *in vivo* depleted of rpL34 (Y2907).** Folding states of rRNA domains and a structure overview are shown for Nog1TAP_L34-A in (A) and (B) and for Nog1TAP_L34-B in (C) and (D). Structure model visualisation and protein colouring for (B) and (D) is described in the legend of Fig 2 and rRNA domain visualisation and colouring for (A) and (C) in the legend of Fig 3. In (A)—(D) the LSU subunit interface side is shown on the left and the subunit solvent surface side on the right. The individual LSU rRNA helices as well as the protein composition and model-based interaction networks detected in structure models Nog1TAP_L34-A and Nog1TAP_L34-B are schematically represented in figures M and N in S1 Appendix.

In summary these structural studies indicated that both rpL25 and rpL34 have a crucial role for the stable positioning of LSU rRNA domain III in nuclear LSU precursors. Moreover, they provided evidence that initial positioning and remodelling of LSU rRNA domains IV and V and the 5S RNP depend on the upstream assembly of rpL25 and rpL34.

## Evidence for a major role of the multi-domain binder rpL2 on nuclear LSU subunit interface formation

Further single molecule cryo-EM analyses revealed a different folding phenotype for Nog1-- TAP associated LSU particles from cells in which expression of rpL2 was shut down. RpL2 is required for the trimming of the ITS2 spacer towards the 5.8S rRNA 3' end. Hence, here LSU pre-rRNA processing is impaired just downstream of the rpL25 and rpL34 dependent initial

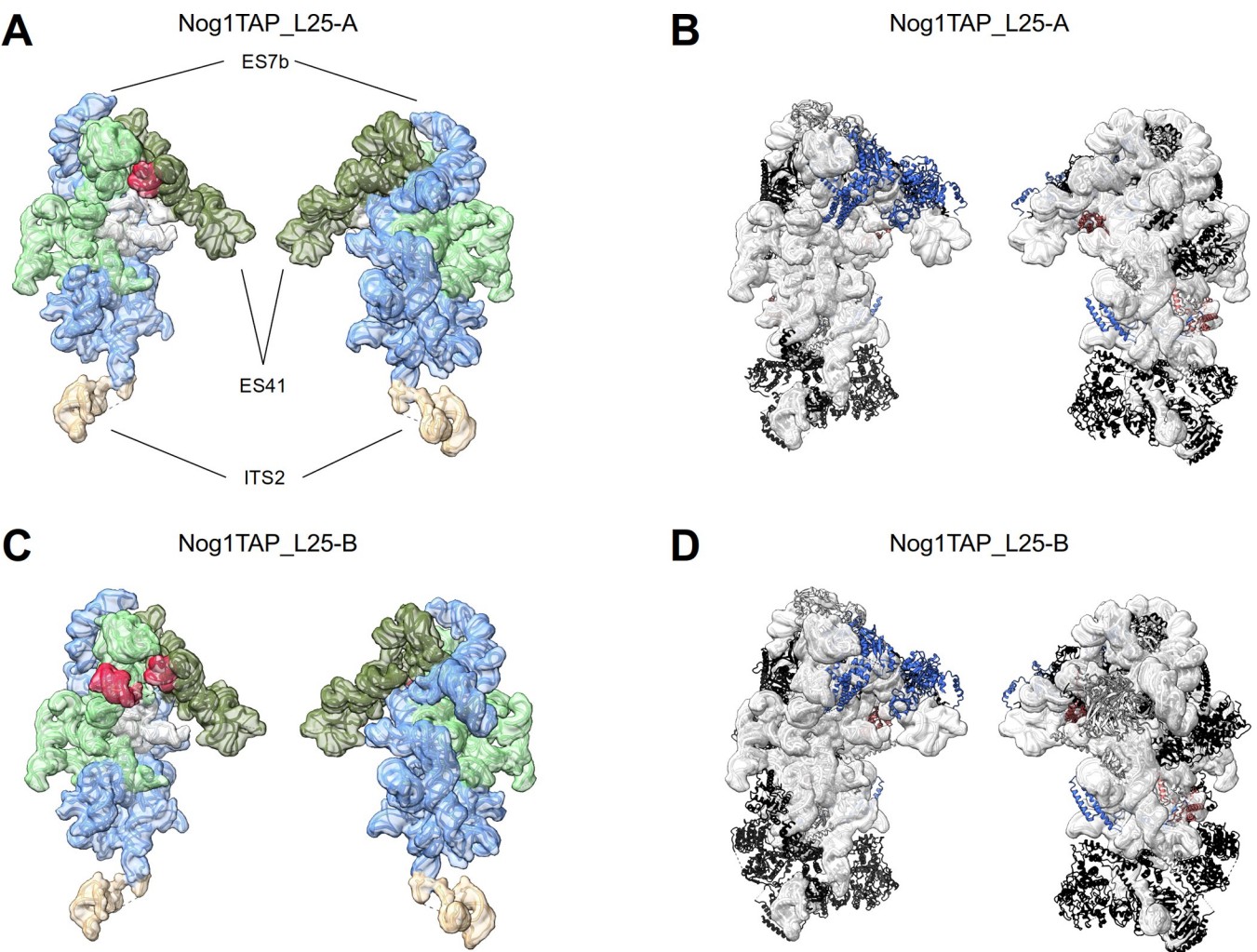

**Fig 5. rRNA folding states and structure models of Nog1-TAP associated particle populations from yeast cells *in vivo* depleted of rpL25 (Y1816).** Folding states of rRNA domains and a structure overview are shown for Nog1TAP_L25-A in (A) and (B) and for Nog1TAP_L25-B in (C) and (D). Structure model visualisation and protein colouring for (B) and (D) is described in the legend of Fig 2 and rRNA domain visualisation and colouring for (A) and (C) in the legend of Fig 3. In (A)—(D) the LSU subunit interface side is shown on the left and the subunit solvent surface side on the right. The individual LSU rRNA helices as well as the protein composition and model-based interaction networks detected in structure models Nog1TAP_L25-A and Nog1TAP_L25-B are schematically represented in figures K and L in S1 Appendix.

cleavage inside the ITS2 spacer. As mentioned above, rpL2 is a multi-domain binder connecting at the subunit interface in a complex with rpL43 the LSU rRNA domains II, III, IV and V.

Three particle populations with dominant folding states, designated here as Nog1TAP_L2-A, Nog1TAP_L2-B and Nog1TAP_L2-C could be distinguished after *in vivo* depletion of rpL2 (see S5 Movie for an animated overview). The rRNA fold of Nog1TAP_L2-B resembled again the one of early intermediate LSU precursors of state Nog1TAP-F (compare Fig 6C and 6D with Figs 2A and 3A, compare Figures I and G in S1 Appendix). Most of the LSU rRNA domains I, II and VI were correctly folded while clear densities attributable to LSU rRNA domains III, IV and V and the 5S rRNA were missing. The ITS2 spacer RNA together with several factors bound to it or to LSU rRNA domains I, II and the ES7 element could not be visualized in Nog1TAP_L2-B.

State Nog1TAP_L2-C (see Fig 6E and 6F and Figure J in S1 Appendix) clearly differed in one point from all the previous folding states of misassembled particles: here, densities could be clearly attributed to the LSU rRNA domain III and several associated factors and r-proteins. Nog1TAP_L2-C thus resembled state Nog1TAP-E of the control sample (compare Fig 6E and 6F with Figs 2B and 3B, compare Figures F and J in S1 Appendix).

Finally, the third population of particles depleted of rpL2 was found in an unprecedented folding state Nog1TAP_L2-A (Fig 6A and 6B, Figure H in S1 Appendix). Here, most of the

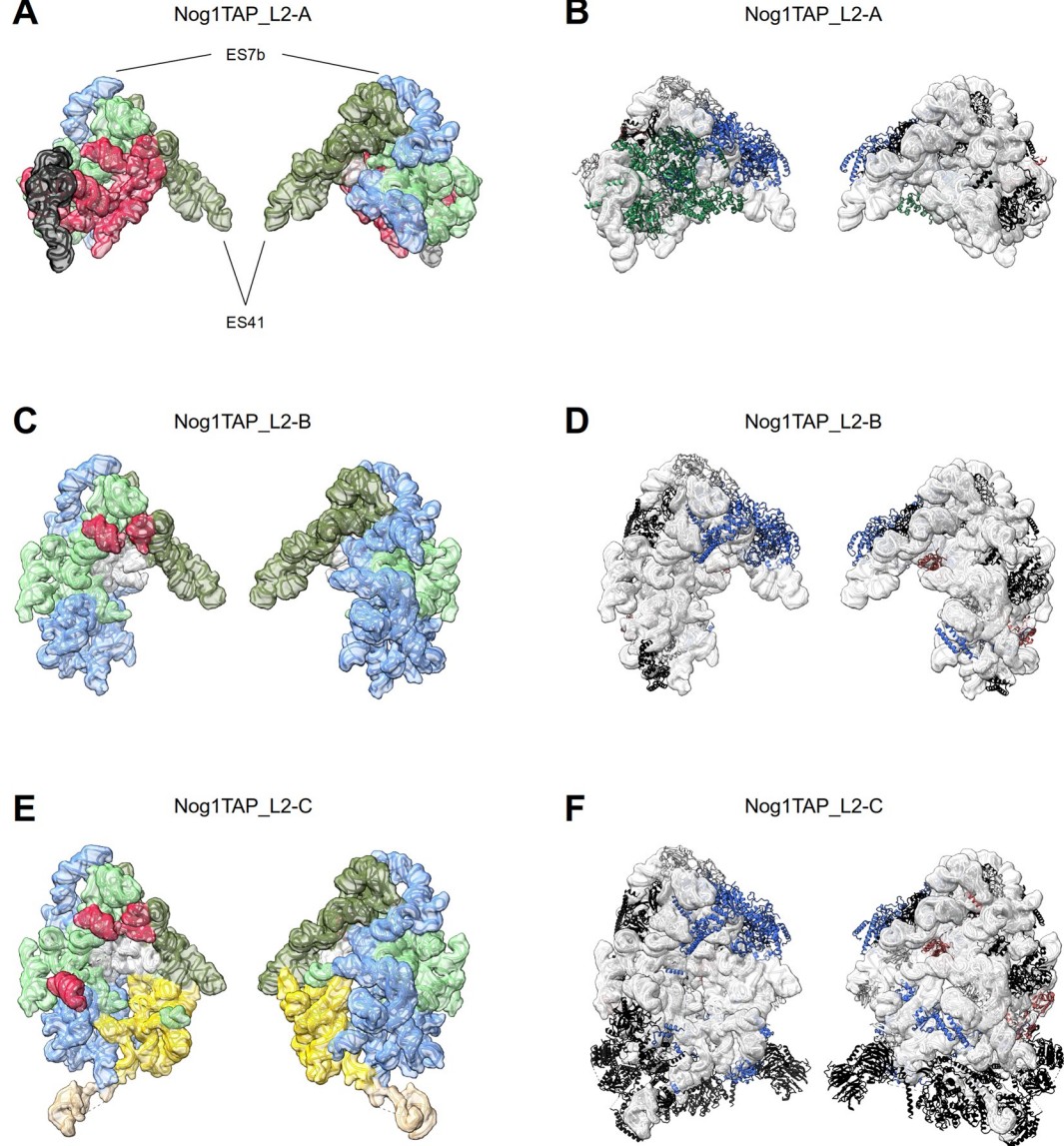

**Fig 6. rRNA folding states and structure models of Nog1-TAP associated particle populations from yeast cells *in vivo* depleted of rpL2 (Y1921).** Folding states of rRNA domains and a structure overview are shown for Nog1TAP_L2-A in (A) and (B), Nog1TAP_L2-B in (C) and (D) and for Nog1TAP_L2-C in (E) and (F). Structure model visualisation and protein colouring for (B), (D) and (F) is described in the legend of Fig 2 and rRNA domain visualisation and colouring for (A), (C) and (E) in the legend of Fig 3. In (A)—(F) the LSU subunit interface side is shown on the left and the subunit solvent surface side on the right. The individual LSU rRNA helices as well as the protein composition and model-based interaction networks detected in structure models Nog1TAP_L2-A, Nog1TAP_L2-B and Nog1TAP_L2-C are schematically represented in figures H, I and J in S1 Appendix.

densities of LSU rRNA domains I, III and IV and the ITS2 spacer were missing. On the other hand, LSU rRNA domain VI was clearly detected and the 5S RNP was docked in the premature rotated position on LSU rRNA domains II and V. These were decorated in a similar way with factors and r-proteins as observed in state Nog1TAP-C from the control sample.

As rpL2 itself and rpL43, most of its major RNA contact sites, including helix H66 in LSU rRNA domain IV and expansion segment ES31 in LSU rRNA domain V were not traceable in any of these states. Still, three of the L2 contact sites were partially visible in Nog1TAP_L2-C. Among them, helix H34 in LSU rRNA domain II and a site between helices H55 and H56 in LSU rRNA domain III were only slightly shifted (see Figure H in S9 Appendix). Helix H75 in LSU rRNA domain V was found still far from its mature position in which it contacts rpL2 after remodelling of the LSU subunit interface.

In sum, the data indicated that initial positioning of LSU rRNA domain III and the 5S RNP can occur independently of rpL2 assembly. By contrast, 5S RNP remodelling and the arrangement of LSU rRNA domain IV and the 5' region of domain V 5' at the subunit interface around the rpL2-rpL43 complex were not detectable.

## Discussion

The structural analyses performed here provide evidence for strong and specific effects on the yeast LSU precursor folding pathway upon blockage of the assembly of three different LSU r-proteins. The respective pre-rRNA processing phenotypes correlated for the utilized yeast mutant strains with the observed LSU folding phenotypes. For the rpL2 mutant strain belonging to pre-rRNA processing phenotype group 3, folding deficiencies in LSU rRNA domains IV and V at the subunit interface and a premature 5S RNP orientation were observed. For the rpL25 and rpL34 mutant strains belonging to pre-rRNA processing phenotype group 2 the LSU folding pathway was compromised upstream of this. Here, the 5S RNP could not be visualized and LSU rRNA domain III was not yet stably positioned in addition to LSU rRNA domains IV and V. Both rpL25 and rpL34 establish significant interactions with LSU rRNA domain III (see Figures A and B in S1 Appendix for an overview). Hence, insufficient LSU rRNA domain III assembly is likely a primary cause for the structural effects observed on this region. Indeed, previous in-depth mutant analyses highlighted the importance of the yeast rpL25 interaction with LSU rRNA domain III for ITS2 cleavage (~ phenotype group 2) and LSU maturation [35,41,42]. They furthermore indicated that changes in the LSU rRNA domain III primary structure can lead to severe delay in ITS2 cleavage independent of their effect on rpL25 binding [43,44].

Compared to rpL34, which is deeply embedded into the LSU rRNA domain III architecture, rpL25 has more extensive contact interfaces to outside components. Several of them, as helix H10 in LSU rRNA domain I, the LSU rRNA domain I binder rpL35 (group 2 phenotype) and the ITS2 binding factor Nop15 can be detected already before the LSU rRNA domain III in LSU precursor density maps (see Figure G in S1 Appendix). It is thus plausible that in case of rpL25 the failure to stably position LSU rRNA domain III might additionally relate to these bridging interactions.

Stable folding and association of LSU rRNA domain III with the preformed domain I-II-VI core particle might affect the downstream arrangement of the other domains IV and V at the subunit interface, and of the 5S RNP in a hierarchical fashion. In line with this, previous biochemical studies revealed that the stable incorporation of several r-proteins and factors depend on LSU rRNA domain III assembly events [24,25]. Among them are the direct rpL25 interactor Nop53 and the r-proteins rpL2 and rpL43 which bind to LSU rRNA domain III (see Figure A in S1 Appendix). Importantly, rpL2 and rpL43 also contact extensively LSU rRNA

domain IV and might therefore contribute to its spatial arrangement in relation to domain III. Indeed, the present analyses of folding states of LSU precursors depleted of rpL2 provided direct evidence for its importance for the positioning of the domain IV and for the related LSU subunit interface remodelling. Hence, effects of LSU rRNA domain III assembly on these aspects of the LSU rRNA folding pathway can be directly attributed to its role for stable recruitment of rpL2 and rpL43.

Initial arrangement of the 5S RNP in its premature position was still detected after *in vivo* depletion of rpL2, although only in a smaller subpopulation of ribosomal particles (~21k particles in Nog1TAP_L2-A of ~98k total ribosomal particles, see Figure B in S5 Appendix) when compared to the control sample (~147k particles in Nog1TAP-A, Nog1TAP-B and Nog1-TAP-C of total ~267k ribosomal particles, see Figure A in S5 Appendix). This indicates that besides stable assembly of rpL2 and its effects on the positioning of LSU rRNA domains IV and V, additional pathways contribute to enable initial 5S RNP positioning downstream of LSU rRNA domain III assembly.

We assume that many more of the r-proteins which are required for efficient LSU rRNA processing, stabilisation and nuclear export are playing important roles for the progression of LSU rRNA domain integration and for their remodelling. In support of this in a previous study the truncation of yeast rpL4 led to perturbations in the LSU folding pathway [34]. In the present work, the strong impact for rpL2, rpL25 and rpL34 assembly on hierarchical folding of several full LSU (sub-)domains provide new insights into their role in subunit stabilisation. On one hand, the 5' end of the 5.8'S rRNA is already protected in the mis-assembled particles from exonucleolytic trimming by multiple parallel acting means [26,45,46]: 1) the 5.8S rRNA 5' end is embedded in a double strand through formation of helix 2, 2) it is bound by rpL17, and 3) its steric access is safely blocked through positioning of LSU rRNA domain VI on top of it. That is consistent with the defined processing state of the 5.8S rRNA 5' end which was previously observed for all the three r-protein assembly mutants [21]. On the other hand, the non-positioned flexible (sub-)domains in the mis-assembled particles should substantially increase at a multitude of other sites the accessibility for general nuclear RNA degradation machineries. That effect is likely still increased in the subpopulations of misassembled particles for which numerous factors were not anymore detectable (Nog1TAP_L2-B, Nog1TAP_L25-A, Nog1-TAP-L34B, see S1 Appendix, compare also with mutant LSU precursor populations depleted of factors in [47]). These populations might represent newly made subunits with limited access to these factors, possibly through a delay of their release from accumulating misassembled particles.

Overall, we consider that the decision to further mature or to degrade a nuclear LSU precursor is primarily under kinetic control, with compact and defined folding states protecting from degradation [48]. Consequently, insufficient r-protein assembly might speed up degradation through the observed blockage at key points of the LSU folding pathway. Strong hierarchical effects on the particle folding states by the assembling components in an environment of overall high RNA decay activity may thus define an intrinsic compositional control mechanism during yeast LSU maturation.

## Materials and methods

### Affinity purification of TAP tagged Nog1 associated particles for cryo-EM analyses

TAP-tagged [49] Nog1 and associated particles were purified from total cellular extracts in one step using rabbit IgG coupled to magnetic beads as described in [50,51] with minor modifications. Yeast strain Y1877 and conditional r-protein expression mutant strains Y1813, Y1816,

Y1921 and Y2907 (described in [24,25]) were cultivated at 30°C in YPG (1% yeast extract, 2% bacto peptone, 2% galactose) followed by incubation in YPD (1% yeast extract, 2% bacto peptone, 2% glucose) for 4 hours at 30°C. Cells from 2 litres of culture (optical densities at 600nm between 0.8 and 1.5) were harvested by centrifugation for 5 minutes at room temperature at 5000g. The cells were washed once in ice cold water and then stored at -20°C. Cells were thawed on ice, washed once in 40 millilitres of buffer A200 (200 mM potassium chloride, 20mM Tris pH8, 5mM magnesium acetate) and were then suspended per gram of wet cell pellet in 1.5 millilitres of buffer A200 with 0.04U/millilitre RNasin (Promega), 1 mM phenyl-methylsulfonyl fluoride and 2 mM benzamidine. The cellular suspension was distributed in portions of 0.8 millilitres to reaction tubes with 2 millilitre volume to which 1.4 grams of glass beads (0.75–1 millimetre, Sigma) were added. Cells were disrupted by shaking them six times for six minutes at full speed on a Vibrax shaking platform (IKA) placed in a room with 4°C ambient temperature. The samples were cooled on ice in between shaking cycles. The crude extract was cleared by two consecutive centrifugation steps for 5 and 10 minutes at 15000g at 4°C. The protein concentration of the resulting supernatant was determined by the Bradford protein assay (Bio-Rad). Extract volumes containing 200 micrograms of protein was added to 500 microlitres cold buffer AE (50mM sodium acetate pH 5.3 and 10mM ethylenediaminetet-raacetic acid) and stored at -20°C for subsequent RNA extraction and northern blotting analysis. The cleared extract was adjusted to 0.5% Triton X100 and 0.1% Tween 20 and added to magnetic beads prepared from 400 microlitres of an IgG (rabbit serum, I5006-100MG, Sigma)-coupled magnetic beads slurry (1 mm BcMag, FC-102, Bioclone) equilibrated in buffer A200. After incubation for one hour at 4°C the beads were washed five times with 1 millilitre cold buffer A200 with 0.5% Triton X-100 and 0.1% Tween 20 and two times with buffer A200. Five percent of the suspension was added to 500 microlitres of ice-cold buffer AE and stored at -20°C for subsequent RNA analyses by northern blotting. The residual magnetic beads were then incubated for two hours at 4°C and for one hour at 16°C in 83 microlitres of buffer A200 with 20 Units RNasin (Promega), 12mM DTT and 16 micrograms of TEV-Protease. The eluate was taken off from the beads and 5% were added to 500 microlitres of ice-cold buffer AE and stored at -20°C for subsequent RNA analyses by northern blotting. 3 microlitres of undiluted eluate and of eluate which was diluted 1:2 or 1:4 in buffer A200 were blotted onto holey carbon R 1.2/1.3 copper 300 mesh grids (Quantifoil) and vitrified in liquid ethane using a Vitrobot MarkIV device (Thermofisher). Blotting and waiting times were 5 seconds and the grids were hydrophilized and cleaned just before sample application using a Pelco-Easiglow system (two times 100 seconds, 0.4mbar, 15mA, air atmosphere). Grids were stored in liquid nitrogen and pre-screened for appropriate particle concentration and ice quality on a JEM-2100F (JEOL) electron microscope equipped with a single-tilt 626 liquid nitrogen cryo-transfer holder (Gatan) and a TEMCAM-F416 camera (TVIPS). Final data acquisition was performed on a Titan-Krios G3 electron microscope equipped with an X-FEG source and a Falcon III direct electron detector camera (Thermofisher) (see S3 Appendix for acquisition parameters). The linear mode of the Falcon III detector was used for data acquisition [52].

## RNA extraction and northern blotting

RNA extraction from samples taken during affinity purification was performed as described in [53] with minor modifications. Samples in buffer AE (see above) were thawed on ice and 500 microlitres of phenol (equilibrated in buffer AE) and 50 microlitres of 10% (w/v) SDS were added. After 6 minutes of vigorous shaking at 65°C the samples were cooled down on ice for two minutes and then centrifuged at 13000g and 4°C for two minutes. The upper layer was transferred to a new reaction tube containing 500 microlitres of

phenol (equilibrated in buffer AE) and after vigorous mixing for 10 seconds the mixture was again centrifuged at 13000g and 4˚C for two minutes. This procedure was repeated once with 500 microlitres of chloroform. The upper layer was again carefully taken off and RNA contained was precipitated by adding 2.5 volumes of ethanol and 1/10 volume of 3M sodium acetate pH5.3. In case of affinity purified fractions, 10 micrograms of glycogen (Invitrogen) were added. Each sample was then briefly mixed and incubated for more than 10 minutes at -20˚C before centrifugation for 30 minutes at 4˚C and 13000g. The supernatant was carefully discarded, and the pellet was dissolved in 20 microlitres of ice-cold water. RNA separation on formaldehyde/MOPS agarose gels (1.3%) or urea/TBE/poly-acrylamide gels (8%) and northern blotting to positively charged nylon membranes (MP Biomedicals) were done essentially as described in [54]. Hybridization with radioactively (32P) end labelled oligonucleotide probes indicated in the figure legends was done overnight in a buffer containing 50% formamide, 1 milligram/millilitre Ficoll, 1 milligram/millilitre polyvinylpyrrolidone and 1 milligram/millilitre bovine serum albumin, 0.5% (w/v) SDS and five times concentrated SSC (twenty times concentrated SSC: 3 M sodium chloride, 300 mM sodium citrate) at 30˚C. The sequence of the oligonucleotide probes used were 5′-CTCCGCTTATTGATATGC-3′ for O212, 5′-GGCCAGCAATTTCAAGTTA-3′ for O210 and 5'-TTTCGCTGCGTTCTTCATC-3' for O209. After hybridization with the probes the membranes were shortly washed once at ambient temperature with two times concentrated SSC and then once for 15 minutes with two times concentrated SSC and once for 15 minutes with SSC at 30˚C. Labelled (pre-)rRNA signals were detected on the washed membranes using a Typhoon Imager FLA9500 (GE Healthcare). For re-probing, membranes were incubated twice with 100 millilitres of a boiling 0.1% (w/v) SDS solution in water. Each time the solution was let cool down to ambient temperature and then discarded. Signals were quantified using ImageJ.

## Cryo-EM data processing and structure model fitting, interpretation and visualization

Cryo-EM data processing was performed in Relion 3.0 [55]. Beam induced motion correction was done using Relions own implementation of the MotionCor2 algorithm [56]. Contrast transfer functions of images were estimated using CTFFIND4.1 [57]. Candidate particles were picked using Relions template based auto-picking algorithm. More details on single particle processing and classification strategies for the different experimental datasets are shown in S5 Appendix. Relions 3D auto-refine algorithm was used with the solvent-flattened FSC option for final refinements with the respective subsets of particle images. For model generation, the maps were further modified using the density-modification procedure in the phenix software package [58,59].

Starting model 3jct was published in [11], starting model 6n8j in [12], and starting models 6elz and 6em1 in [9]. The model of *S. cerevisiae* Has1 was taken from pdb 6c0f [10]. Initial rigid body fitting and model editing was done in UCSF Chimera [60] and in UCSF ChimeraX [61]. The fit of the models to the respective experimental maps was further improved by molecular dynamics flexible fitting using the Isolde plugin in UCSF ChimeraX [62]. Molecular dynamics flexible fitting was first done for individual rRNA domains and their surrounding with distance restraints applied between nucleobases (typically distance cutoff 5 and kappa 100) which were then gradually relaxed. Final flexible fitting included the complete model. Global model geometry parameters (including nucleic acids geometry and rotamer and ramachandran outliers) were regularized using the geometry minimization tool in the phenix software packet. Map and model statistics shown in S3 Appendix were obtained from Relion

(resolution) and from the comprehensive cryo-EM validation tool in the phenix software package [59]. Model statistics shown in S6 and S7 Appendixes were obtained in UCSF ChimeraX using Python scripts. Here, possible direct residue interactions were predicted using the ChimeraX command "contacts" with default settings for overlapCutoff and hbondAllowance parameters. Python scripts in UCSF ChimeraX were also used to generate the model representations in Figs 2–6, the diagrams indicating coverage in models of individual rRNA helices in S1 Appendix and for data export to the Cytoscape software packet [63]. The latter was used for the schematic visualisation of protein components and their predicted major interaction interfaces shown in S1 Appendix. S1–S5 Movies were generated with scripting commands in ChimeraX. Yeast LSU rRNA domain and helices definitions were taken from [7]. All scripts are available upon request. The model of the mature large ribosomal subunit represented in Figs 2 and 3, in S1 Appendix and in S1 and S2 Movies was taken from pdb entry 4u3u [64].

## Supporting information

**S1 Appendix. Schematic representation of rRNA helices, protein composition and predicted major interaction interfaces in the mature LSU and in single particle cryo-EM derived structure models.** Individual proteins are represented in (A)–(N) by rounded corners, groups of proteins by rectangles with sharp corners. For space reasons throughout the Figure rRNA domains are indicated with Arabic numbering (D0, D1, D2, D3...). Definition of multi-component r-protein clusters designated Cl_D1-D5, Cl_D1-D2, Cl_D2-D6 and Cluster Cl_D3 and Cl_L2-L43 is shown in (A) and is based on protein—protein interactions and functional categories. These are symbolized in (A)–(N) for all proteins by the colour of the border of the rectangles: Proteins required for early pre-rRNA processing are represented by black bordered rectangles, for intermediate nuclear pre-rRNA processing steps by blue, for late nuclear pre-rRNA processing steps by green and for downstream nuclear maturation and export by red bordered rectangles. Boxes for proteins with unclear function and non-essential proteins have grey borders, with the latter ones in dotted lines (see also S6 Appendix for definition of functional categories). Interactions between individual proteins, groups of proteins and rRNA domains were deduced from residue—residue proximities in the structure models (see Materials and Methods). Predicted major interaction interfaces with $> = 10$ residues involved (respective numbers are shown, sums of residues in both partners) are visualized between r-protein (groups) by grey lines, between (groups of) factors by white lines and between factors and r-proteins by interrupted lines. For r-protein interactions with r-proteins also the ones involving less than 10 residues are visualized and the number behind the slash is the one observed in the mature LSU structure model 4u3u. Numbers falling below 50% of the ones deduced for pdb-databank entry 4u3u are highlighted in red. Predicted interactions of proteins with rRNA domains are shown as bar diagrams inside protein boxes (relative amounts). Colour code and order of rRNA domains in these diagrams are reflected in the bar diagram at the bottom of each figure. Here, the percent of residues of each rRNA helix which was observed in the respective model is represented. More details on residues modelled for individual proteins and RNA helices and their predicted interactions can be found in S6 Appendix (proteins) and S7 Appendix (RNA helices).
(PDF)

**S2 Appendix. Growth of yeast conditional r-protein expression mutants on galactose and glucose containing medium.** Yeast strains BY4742 (lane 1), Y1877 (lane 2), Y1816 (lane 3), Y1921 (lane 4) and Y2907 (lane 5) were cultivated in galactose containing liquid full medium at 30˚C and serial dilutions were then spotted on galactose (YPG) or glucose containing

(YPD) solid medium. Images were taken after 72h incubation at 30˚C.
(PDF)

**S3 Appendix. Cryo-EM data collection parameters, map resolutions, accession numbers and model parameters for cryo-em based structure models described in this work.** Map resolution estimates (half maps, fourier shell correlation threshold 0.143) are given as reported by relion (user-created mask) and by phenix validation tools (no user-created mask). Model statistics as reported by phenix validation tools and accession numbers for models (wwPDB), related density maps (EMDB) and for full EM-datasets (EMPIAR) are indicated. Fourier shell correlation (FSC) graphs as reported by phenix validation tools are shown in S4 Appendix.
(PDF)

**S4 Appendix. FSC graphs for cryo-EM based density maps analyzed in this work as reported by phenix validation tools.**
(PDF)

**S5 Appendix. Single particle sorting and processing schemes for cryo-EM datasets recorded in this study.** White numbers in 3D classification views indicate particle counts in the respective classes. Data processing schemes are shown for Nog1-TAP associated particles from strain Y1877 in (A), strain Y1921 in (B), strain Y1816 in (C) and Y2907 in (D). All shown intermediate and final density maps obtained by Relion's 3D-Autorefine procedure are represented using the same dimensions and orientation.
(PDF)

**S6 Appendix. Structure models: Protein data.** Excel table with chain and protein names, modelled residues and model-data based predictions for interactions with other (groups of) proteins, rRNA domains and individual rRNA helices for each of the described models and for the LSU in pdb entry 4u3u.The number of residues involved in predicted interactions are shown in brackets (unidirectional). Literature based functional classification for modelled proteins are given in the Excel table's tab named "Functional categories".
(XLSX)

**S7 Appendix. Structure models: RNA data.** Excel table listing for all models and for the LSU in pdb entry 4u3u for each LSU rRNA helix the modeled residues and the model-data based predictions for interactions with (groups of) proteins, rRNA domains and individual rRNA helices (unidirectional). The number of residues involved in predicted interactions are shown in brackets.
(XLSX)

**S8 Appendix. Quantitation of northern blotting experiments shown in Fig 1.**
(PDF)

**S9 Appendix. Snapshots of single particle cryo-EM derived structure models.** The indicated structure models were visualized in ChimeraX with RNA shown in cartoon backbone representation and proteins in surface representation, except for G) and H) where proteins are also shown in cartoon backbone representation. In G) and H) models obtained from r-protein expression mutant strains were aligned by the matchmaker algorithm in ChimeraX to the indicated reference models based on the 25S rRNA chains.
(PDF)

**S1 Movie. Positions of r-proteins studied in this work in mature ribosomes.** rRNA atoms are shown as spheres following the colouring scheme for rRNA domains used in Figs 3–6 and in S1 Appendix. The indicated r-proteins are shown in white in cartoon backbone

representation.
(MP4)

**S2 Movie. rRNA folding states of Nog1-TAP associated LSU precursor particles from yeast cells with endogenous r-protein expression levels.** rRNA atoms are shown as spheres following the colouring scheme for rRNA domains used in Figs 3–6 and in S1 Appendix. Names of the structure models visualized are shown on top. The relative proportion of particles (of total candidate ribosomal particles identified in the sample) used to calculate densities was 36% (Nog1TAP-A), 7% (Nog1TAP-B), 13% (Noh1TAP-C), 4% (Nog1TAP-E) and 3% (Nog1-TAP-F).
(MP4)

**S3 Movie. Folding states of Nog1-TAP associated LSU precursor particles from yeast cells depleted of rpL34.** rRNA atoms are shown as spheres following the colouring scheme for rRNA domains used in Figs 3–6 and in S1 Appendix. R-proteins are shown in wheat and ribosome biogenesis factors in green colour as indicated. Names of the structure models visualized are shown on top. The relative proportion of particles (of total candidate ribosomal particles identified in the sample) used to calculate densities was 46% (Nog1TAP_L34-A) and 11% (Nog1TAP_L34-B).
(MP4)

**S4 Movie. Folding states of Nog1-TAP associated LSU precursor particles from yeast cells depleted of rpL25.** rRNA atoms are shown as spheres following the colouring scheme for rRNA domains used in Figs 3–6 and in S1 Appendix. R-proteins are shown in wheat and ribosome biogenesis factors in green colour as indicated. Names of the structure models visualized are shown on top. The relative proportion of particles (of total candidate ribosomal particles identified in the sample) used to calculate densities was 61% (Nog1TAP_L25-A) and 30% (Nog1TAP_L25-B).
(MP4)

**S5 Movie. Folding states of Nog1-TAP associated LSU precursor particles from yeast cells depleted of rpL2.** rRNA atoms are shown as spheres following the colouring scheme for rRNA domains used in Figs 3–6 and in S1 Appendix. R-proteins are shown in wheat and ribosome biogenesis factors in green colour as indicated. Names of the structure models visualized are shown on top. The relative proportion of particles (of total candidate ribosomal particles identified in the sample) used to calculate densities was 22% (Nog1TAP_L2-A), 49% (Nog1-TAP_L2-B) and 20% (Nog1TAP_L2-C).
(MP4)

**S1 Raw images.**
(PDF)

## Acknowledgments

We thank Ralph Witzgall (Chair of Molecular and Cellular Anatomy, University of Regensburg) for providing access to the JEM-2100F and Reinhard Rachel (Centre for Electron Microscopy, University of Regensburg) for his support in operating the JEM-2100F. We thank Lifei Fu (Chair of Biophysics II, University of Regensburg) and Norbert Eichner and Gerhard Lehmann (both Chair of Biochemistry I, University of Regensburg) for their help during setup of a Relion GPU workstation. We are grateful to Christoph Engel (Structural Biochemistry, University of Regensburg) for his continuous support and willingness to share his structural

biological advice. Cryo-EM image acquisition at the Titan Krios was supported by Dr. Bettina Böttcher and Christian Kraft in the cryo-EM-facility of the Julius-Maximilian University Würzburg. We thank the ChimeraX-team at the RBVI for their support in Python scripting in UCSF ChimeraX, developed by the Resource for Biocomputing, Visualization, and Informatics at the University of California, San Francisco.

## Author Contributions

**Conceptualization:** Philipp Milkereit.

**Funding acquisition:** Joachim Griesenbeck, Herbert Tschochner, Philipp Milkereit.

**Investigation:** Gisela Pöll, Philipp Milkereit.

**Methodology:** Gisela Pöll, Michael Pilsl.

**Project administration:** Joachim Griesenbeck, Herbert Tschochner, Philipp Milkereit.

**Software:** Philipp Milkereit.

**Supervision:** Philipp Milkereit.

**Visualization:** Philipp Milkereit.

**Writing – original draft:** Philipp Milkereit.

**Writing – review & editing:** Gisela Pöll, Michael Pilsl, Joachim Griesenbeck, Herbert Tschochner.

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
