## [Decision Letter · Decision Letter 0]

22 Jun 2021

PONE-D-21-15939

Analysis of subunit folding contribution of three yeast large ribosomal subunit proteins required for stabilisation and processing of intermediate nuclear rRNA precursors.

PLOS ONE

Dear Dr. Milkereit,

Thank you for submitting your manuscript to PLOS ONE. After careful consideration, we feel that it has merit but does not fully meet PLOS ONE’s publication criteria as it currently stands. Therefore, we invite you to submit a revised version of the manuscript that addresses the points raised during the review process.

The reviewers all had suggestions for minor revision, which you should follow. Of particular note, all three commented on the difficulty of understanding your figures. This should be addressed with priority.

We look forward to receiving your revised manuscript.

Kind regards,

Thomas Preiss, PhD

Academic Editor

PLOS ONE

Journal Requirements:

"We are grateful to Christoph Engel (Structural Biochemistry, University of Regensburg) for his

continuous support and willingness to share his structural biological advice. Cryo-EM

image acquisition at the Titan Krios was supported by Christian Kraft in the cryo-EM635

facility of the Julius-Maximilian University Würzburg (INST 92/903-1FUGG). We

thank the ChimeraX-team at the RBVI for their support in Python scripting in UCSF

ChimeraX, developed by the Resource for Biocomputing, Visualization, and

Informatics at the University of California, San Francisco, with support from National

Institutes of Health R01-GM129325 and the Office of Cyber Infrastructure and

Computational Biology, National Institute of Allergy and Infectious Diseases."

"This work was supported by a grant which was given in the collaborative research

center SFB 960, from the "Deutsche Forschungsgemeinschaft" (www.dfg.de) to JG, HT

and PM.

The funders had no role in study design, data collection and analysis, decision topublish, or preparation of the manuscript."

Reviewers' comments:

Reviewer's Responses to Questions

**Comments to the Author**

1. Is the manuscript technically sound, and do the data support the conclusions?

Reviewer #1: Yes

Reviewer #2: Yes

Reviewer #3: Partly

2. Has the statistical analysis been performed appropriately and rigorously? 

Reviewer #1: N/A

Reviewer #2: I Don't Know

Reviewer #3: Yes

3. Have the authors made all data underlying the findings in their manuscript fully available?

Reviewer #1: Yes

Reviewer #2: Yes

Reviewer #3: Yes

4. Is the manuscript presented in an intelligible fashion and written in standard English?

Reviewer #1: Yes

Reviewer #2: Yes

Reviewer #3: Yes

5. Review Comments to the Author

Reviewer #1: This is an excellent study analyzing structural changes of LSU precursors upon depletion of three selected large ribosomal subunit proteins. The manuscript is well written and the results add important new knowledge to the field. Still, some improvements are necessary to enhance the readability of the paper.

- My main concern is the presentation of the Figures. None of the proteins or RNA domains visible in the structures are labeled. Therefore, it is difficult to follow the descriptions in the text. It cannot be expected that all readers are familiar enough with the structures to recognize all the elements described in the text without labels in the Figures.

- Moreover, the names of the shown particel populations are counter-intuitive. The first population that is discussed is Nog1-TAP-F, which is shown in Figures 2A and 3A. Population E is shown in Figures 2B and 3B, etc. If the authors want to stick with this nomenclature, the names of the classes should also be indicated directly in the Figures.

- Figures 2F and 3F are not mentioned in the text

- Some kind of summary Figure in the end would be useful. E.g. an overview of the classes found in WT, with percentage of representation relative to the total number of particles for each class, and an indication which of these classes (or similar structures) are found upon depletion of the individual RPL proteins (including percentage relative to total amount of particles in the respective purifications)

- The abstract is very general. The abstract would be more informative if at least the names of the three proteins whose role in subunit folding was studied were mentioned.

- Figure Legend S1: the text for (B) is missing

Reviewer #2: Review for Manuscript ID PONE-D-21-15939 entitled " Analysis of subunit folding contribution of three yeast large ribosomal subunit proteins required for stabilisation and processing of intermediate nuclear rRNA precursors."

Functional coupling of r-protein assembly with the stabilization and maturation of subunit precursors potentially promotes the cellular production of ribosomes. However, the precise mechanism remains exclusive. In this study, the authors tried to decipher an intrinsic quality control pathway by the evaluation of the contribution of three yeast large ribosomal subunit r-proteins, rpL2, rpL25, rpL34, for intermediate nuclear subunit folding steps. They found that a strong impact for rpL2, rpL25, and rpL34 assembly on hierarchical folding of several full LSU (sub-)domains that provide new insights into their role in subunit stabilization. In the rpL2 mutant strain, they observed folding deficiencies in LSU rRNA domains IV and V at the subunit interface and a premature 5S RNP orientation. The folding states of LSU precursors depleted of rpL2 provided direct evidence for its importance for the positioning of the domain IV and for the related LSU subunit interface remodeling. In the rpL25 and rpL34 mutant strains, they observed the 5S RNP could not be visualized and LSU rRNA domain III was not yet stably positioned in addition to LSU rRNA domains IV and V.

Based on these observations, they propose that insufficient r-protein assembly might speed up degradation through the observed blockage at key points of the LSU folding pathway. Their proposal will be of broad interest and potentially interesting. However, appropriate revision is needed before publication in PLOS ONE. They give appropriate credits to previous work.

Comments:

1. The authors should demonstrate the Figures in a more appropriate way for general readers. More information related to three rpL proteins should be included. For instance, it is better to demonstrate the position of three rpL proteins in Figure 2. In addition, the superimposing of the WT and the rpL-depleted structures could clarify their differences.

Reviewer #3: The authors of the paper entitle “Analysis of subunit folding contribution of three yeast large ribosomal subunit proteins required for stabilisation and processing of intermediate nuclear rRNA precursors” studied the effects of three r-proteins rpL2, rpL25 and rpL34 on the maturation of yeast ribosomes by analyzing post-transcriptional processing of ribosomal RNA and structures obtained by cryo-EM of several ribo-particles precursors of ribosome maturation. The authors discovered that the absence of expression of either the rpL25 or rpL34 affect the post-transcriptional cleavage of the internal spacer that connect the 5.8S rRNA with the 28S rRNA. These effects are interconnected with the absence of structuration of the 28S rRNA domain III. On the other hand, the absence of expression of the rpL2 protein affect trimming of the 7S RNA which is necessary to generate a mature 5.8S rRNA. Also, the absence of the rpL2 protein affect structuration of the domains IV and V as expected. Their results suggest possible steps in the maturation of eukaryotic ribosomes where the rpL34 and rpL25 are needed to structure domain III which will stablish maturation of the pre-rRNA and posterior interactions of the rpL2 protein required for further maturation of the newly formed 5.8S rRNA and domains IV and V.

General observations

This study had got very exciting structures that would complement previous work on ribosome maturation. However, this reviewer had hard time following the results’ descriptions with the provided figures, mostly because there is no enough information in the figures that can help the reader to follow the text of the document. Also, there are some questions about the results that this reviewer considers important to discuss.

1) Supplementary Figure 1. This figure is important for the paper and very complex which legend do not have enough information for each of its panels. Unfortunately, the authors use the nomination D1-7 that is confusing with the rest of the paper. The authors should consider sticking with the usage of DI-DVII. Also, I am wondering if using the secondary structure of the 28S rRNA/5.8S rRNA would be better option to color code the interaction sites of the r-Proteins.

2) What do the authors mean with “Production of ribosomes with defined composition”? Also, I do not see at the discussion section anything about this term.

3) I could not find any reason of why the authors decided to use these three proteins and not others? The paper would benefit from the addition of a sentence of two explaining the reasoning of using these proteins, perhaps using as a reference supplementary figure 1.

4) Similar to question 3, I could not find any reasoning on why the authors decided to use Nog1 as an anchor to pull out ribonucleoproteins? Also, supplementary figure 2 do not show such nog1-Tap growth delay as stated in the text. I am wondering if I am missing something. Size of the colony?

5) On materials and methods. For how long the cultures grew under the presence of galactose? I agree that reduction of growth could be a good indication of the absence of the studied r-proteins. However, I am wondering if the cells are really lacking of these proteins in the selected period of growth. Did the authors performed western blots to follow the presence of the studied r-proteins after shifting the culture to glucose ? Also, I am wondering about the type of ribonucleoprotein materials that were present in the cells during the purification procedure. Did the authors analyzed polysome profilings before and after turning down the expression of the studied r-proteins? Perhaps they are losing information by pulling out just specific complexes attached to Nog1.

6) About Figure 1. Consider to add densitometry calculations of each band to support your statements in the text. Also, why is 5.8S observed at the rpL34 and rpL25 samples if 7S was not produced? Is 5.8S very stable and perhaps coming from previous mature ribosomes? Why is that at the rpL2 sample the 25.5S signal is not as abundant as the control when the 7S is?

7) Figures 2 and 3 needs extra labels. Please consider to indicate the location of the 5S rRNA, rpL2, rpL25, rpL34, and Nog1. Also, consider to mark the most important differences between structures.

6. PLOS authors have the option to publish the peer review history of their article (what does this mean?). If published, this will include your full peer review and any attached files.

Reviewer #1: No

Reviewer #2: No

Reviewer #3: No

---

## [Author Response · Author response to Decision Letter 0]

3 Sep 2021

Dear Dr. Preiss,

Thank you for taking the time to serve as scientific editor for our submitted manuscript.

We are also very grateful to the three reviewers for carefully going through the manuscript and for their constructive reviews! 

We agree with the reviewers that any means to facilitate access of readers to the obtained twelve complex structure models besides the two-dimensional overviews in Figs 2-6 and the twelve interaction maps in S1 Appendix would be beneficial. To tackle this issue, we followed the advice of the reviewers by adding further labels for 5S rRNA, Nog1, the rpl2/rpl43 heterodimer and rpL25 in the reference structures of control samples shown in Figs 2-3. As suggested, the names of the structure models are now also directly shown in Figs 2-6.

Secondly, we created in total five additional movies each showing animations of some of the structure models together with respective legends: The first one shows the positions of the three studied r-proteins with respect to the LSU rRNA domains in mature ribosomes (S1 Movie). The second one visualizes the hypothetical model-derived folding-path of rRNA domains during nuclear LSU maturation in control cells (S2 Movie). The next three movies show the observed folding states of immature ribosomal particles obtained from cells depleted of either rpL34, rpL25 or rpL2 (S3, S4 and S5 Movie). Structures shown in the S2-S5 Movies can be directly compared with each other. This should further facilitate tracing of the observed major differences independent of any structure model visualization software. 

For readers which want to directly analyze the structure models and electron density files these are hold for release until publication in the wwwpdb and the EMDB database. The respective accession codes are indicated in S3 Appendix. The original electron microscopy micrographs were uploaded on the EMPIAR database and can be accessed with the updated accession codes provided in S3 Appendix. 

Besides, we added as suggested by reviewer #3 the results of quantitation of the northern blots shown in Fig 1 as supplementary information S8 Appendix. In Fig 1 uncropped regions are shown with complete lanes of the northern blots developed with the indicated probes. We assume that requirements of PlosOne for blot and gel representations should thus be fulfilled.

Finally, we updated the manuscript to include references to the newly added or changed Figures and we deleted as requested the funding information previously given in the Acknowledgement section of the manuscript.

In the following we provide a detailed point by point response to the issues raised by the three reviewers.

Reviewer #1

- My main concern is the presentation of the Figures. None of the proteins or RNA domains visible in the structures are labeled. Therefore, it is difficult to follow the descriptions in the text. It cannot be expected that all readers are familiar enough with the structures to recognize all the elements described in the text without labels in the Figures.

As explained above, we added labels for 5S rRNA, Nog1, the rpl2/rpl43 heterodimer and rpL25 to the reference structures in Figs 2-3 in addition to the previous color coding and labels according to the reviewer’s suggestions. This should further facilitate comprehension of the complicated structure models for readers not closely familiar with these ribosomal structures. In addition, we added five movies showing animations of the structure models together with respective legends to better visualize their three-dimensional organisation. Reference to these movies was added at the relevant locations in the main text.

- Moreover, the names of the shown particel populations are counter-intuitive. The first population that is discussed is Nog1-TAP-F, which is shown in Figures 2A and 3A. Population E is shown in Figures 2B and 3B, etc. If the authors want to stick with this nomenclature, the names of the classes should also be indicated directly in the Figures.

To avoid any confusion, we now show directly in Figs 2-6 the respective names of the structures as used in the text. 

- Figures 2F and 3F are not mentioned in the text

Reference to the mature ribosomal models in Figs 2 and 3 F was now added in the main text.

- Some kind of summary Figure in the end would be useful. E.g. an overview of the classes found in WT, with percentage of representation relative to the total number of particles for each class, and an indication which of these classes (or similar structures) are found upon depletion of the individual RPL proteins (including percentage relative to total amount of particles in the respective purifications)

We think that the newly added S2-5 Movies should provide a decent overview of folding states and their respective hypothetical transitions in the control cells (S2 Movie) and of residual folding states found in the mutant strains (S3-5 Movies). Camera perspectives, animations and legends were maintained in a way to allow for direct comparison of the structures shown in the movies. We indicated the percent of particles (of all identified candidate ribosomal particles in the sample) used for calculation of densities of the respective particle population in the Legends of the new Figures. 

- The abstract is very general. The abstract would be more informative if at least the names of the three proteins whose role in subunit folding was studied were mentioned.

As suggested, we added in the abstract the names of the three r-proteins studied in this work.

- Figure Legend S1: the text for (B) is missing

We corrected the Figure Legend accordingly.

Reviewer #2

- The authors should demonstrate the Figures in a more appropriate way for general readers. More information related to three rpL proteins should be included. For instance, it is better to demonstrate the position of three rpL proteins in Figure 2. In addition, the superimposing of the WT and the rpL-depleted structures could clarify their differences.

As explained in response to reviewer #1:

“We added labels for 5S rRNA, Nog1, the rpl2/rpl43 heterodimer and rpL25 to the reference structures in Figs 2-3 in addition to the previous colour coding and labels according to the reviewer’s suggestions. This should further facilitate comprehension of the complicated structure models for readers not closely familiar with these ribosomal structures. In addition, we added five movies showing animations of the structure models together with respective legends to better visualize their three-dimensional organization. Reference to these movies was added at the relevant locations in the main text. Camera perspectives, animations and legends were maintained in a way to allow for direct comparison of the structures shown in the movies.”

Reviewer #3

- This study had got very exciting structures that would complement previous work on ribosome maturation. However, this reviewer had hard time following the results’ descriptions with the provided figures, mostly because there is no enough information in the figures that can help the reader to follow the text of the document.

As explained in response to reviewer #1:

“We added labels for 5S rRNA, Nog1, the rpl2/rpl43 heterodimer and rpL25 to the reference structures in Figs 2-3 in addition to the previous color coding and labels according to the reviewer’s suggestions. In addition, we added five movies showing animations of the structure models together with respective legends to better visualize their three-dimensional organization. Reference to these movies was added at the relevant locations in the main text. Camera perspectives, animations and legends were maintained in a way to allow for direct comparison of the structures shown in the movies.”

- Supplementary Figure 1. This figure is important for the paper and very complex which legend do not have enough information for each of its panels. Unfortunately, the authors use the nomination D1-7 that is confusing with the rest of the paper. The authors should consider sticking with the usage of DI-DVII. Also, I am wondering if using the secondary structure of the 28S rRNA/5.8S rRNA would be better option to color code the interaction sites of the r-Proteins.

It is true that we introduced here an inconsistency in the terminology for the LSU rRNA domains and we forgot to mention this in the Figure Legend of S1 Appendix. This inconsistency was introduced due to the space limitations we faced when trying to summarize the proteins, rRNA elements and their interactions in the twelve obtained highly complex structure models in as simple as possible two-dimensional maps with their general layout maintained for all models. In terms of space, Arabic numbering for LSU rRNA domains gave some advantage over the original roman one. For this reason, we still would like to keep the numbering, and we explicitly mention this point in the new version of the manuscript in the Figure Legends of S1 Appendix.

Concerning the color coding, the bars of the small diagram-inserts in the protein boxes represent relative amounts of interactions with rRNA domains and are colored according to the color code used for the rRNA domain bar-diagram at the bottom of the Figures. 

- What do the authors mean with “Production of ribosomes with defined composition”? Also, I do not see at the discussion section anything about this term.

We used the term “..promotes the production of ribosomes with defined composition” referring to the idea that many, most or even all newly made cytoplasmic ribosomes in yeast might share a common set of RNA’s (25S rRNA, 5.8S rRNA,..) and r-proteins (rpL1, either rpL2A or rpL2B, rpL3,...) in a defined ration (1:1:1…). We do not use exactly this term again in the discussion section of the manuscript. Still, we discuss there that the strong and hierarchical effects on rRNA folding upon lack of individual r-proteins might promote the degradation of incomplete particles and use in this context for example the term “compositional control mechanism” in the last paragraph of the discussion section. 

- I could not find any reason of why the authors decided to use these three proteins and not others? The paper would benefit from the addition of a sentence of two explaining the reasoning of using these proteins, perhaps using as a reference supplementary figure 1.

We were originally interested in r-proteins which possibly affect yeast intermediate nuclear LSU folding steps, mainly because studies in recent years showed that these particle populations are structurally well resolved in single particle cryo-EM (see introduction, second paragraph). We speculated that r-proteins with known effects on intermediate to late nuclear pre-rRNA processing steps might play a role for these folding steps (r-protein phenotype groups 2 and 3, see introduction, third paragraph). We decided then to choose among them for comparison three r-proteins with different rRNA-domain interconnectivity level (rpL34 ~ 1 domain, rpL25 ~ 2 domains, rpL2 ~ multi-domain), rRNA processing phenotypes (rpL25 + rpL34 ~ phenotype group 2, rpL2 ~ phenotype group 3) and assembly behaviour (rpL2 is in later LSU precursors structural resolvable and less tightly bound to earlier ones). The latter aspects are described in the fourth paragraph of the introduction. To express the comparative concept of this study more clearly, we added an appropriate sentence to the fourth paragraph of the introduction. In addition, we refer there now to the new S1 Movie file in which binding sites of the three studied r-proteins in mature ribosomes are visualized which allows to compare the way how they establish interactions between and in rRNA domains.

- Similar to question 3, I could not find any reasoning on why the authors decided to use Nog1 as an anchor to pull out ribonucleoproteins? Also, supplementary figure 2 do not show such nog1-Tap growth delay as stated in the text. I am wondering if I am missing something. Size of the colony?

We added to the first paragraph of the results a sentence in which we explain that Nog1 was previously found to be part of a wide range of intermediate nuclear LSU precursor populations and that its association with LSU precursors was independent of the ongoing expression of rpL2, rpL25 and rpL34. These were arguments for us to choose Nog1 as bait protein. Concerning the growth delay, yes, the smaller colony size reflects in this experiment the stated growth delay. We added this statement to second paragraph of the results part. 

- On materials and methods. For how long the cultures grew under the presence of galactose? I agree that reduction of growth could be a good indication of the absence of the studied r-proteins. However, I am wondering if the cells are really lacking of these proteins in the selected period of growth. Did the authors performed western blots to follow the presence of the studied r-proteins after shifting the culture to glucose ? Also, I am wondering about the type of ribonucleoprotein materials that were present in the cells during the purification procedure. Did the authors analyzed polysome profilings before and after turning down the expression of the studied r-proteins? Perhaps they are losing information by pulling out just specific complexes attached to Nog1.

All expression mutant strains were continuously grown in galactose containing medium for their cultivation. We previously observed for most of the analysed yeast pGAL1/10 driven ribosomal protein gene mutants, including the ones studied here, strong pre-rRNA processing phenotypes starting at around 2 hours depletion time in glucose containing medium (see as example Pöll et al. doi: 10.1371/journal.pone.0008249, Ferreira-Cerca et al. doi: 10.1016/j.molcel.2005.09.005.). As stated in Materials and Methods, expression shut down in this study was done by incubation for four hours in glucose containing medium. Large quantities of mature ribosomes (>150.000) are observed in logarithmic growing yeast cells and, in agreement with their long half-life, substantial amounts of them can still be detected in cells after these and longer depletion times (see for example input rRNA levels after four hour depletion for rpL2 and rpL25 in Ohmayer et al., doi: 10.1371/journal.pone.0068412, some examples for polysome analyses are shown in Pöll et al. doi: 10.1371/journal.pone.0008249). The early onset of maturation phenotypes is consistent with the stoichiometric need for most of the ribosomal proteins during ongoing rRNA synthesis. This high demand can apparently neither be sufficiently supplied by recycling of the still existing pool of “old” ribosomal proteins which is incorporated in the still existing “old” ribosomes, nor by the residual low expression of respective r-proteins under control of the glucose-repressed GAL1/10 promoter. 

Regarding the last point mentioned by reviewer #3, we agree with the possibility that we still might lack some information here by choosing Nog1 as bait for purification. Particles associated with Nog1 might be folded in a way not compatible with the SPA approach used in this work, and ribosomal sub-populations might be missing due to the choice of the bait. Anyhow, we believe that comparing the results from samples of control cells and the three mutant strains using the very same bait protein and experimental approach, provided strong evidence for the observed significant differences in the respective folding pathways. 

- About Figure 1. Consider to add densitometry calculations of each band to support your statements in the text. Also, why is 5.8S observed at the rpL34 and rpL25 samples if 7S was not produced? Is 5.8S very stable and perhaps coming from previous mature ribosomes? Why is that at the rpL2 sample the 25.5S signal is not as abundant as the control when the 7S is?

We added the quantification of these data now as S8 Appendix and refer to it at the appropriate place in the first paragraph of the result section. 

Regarding the 5.8S rRNA: yes, we consider this as the background level of mature ribosomes which pre-existed before in vivo depletion of the respective r-proteins started. An alternative explanation is that maturation of pre-rRNAs is strongly delayed at these points but not fully blocked allowing for some production of Nog1-TAP associated particles with matured pre-rRNA over time.

In regard to the rRNA composition of the rpl2 sample: In samples of control cells (as after depletion of rpL21) comparably high levels of 25.5+25S, but also of 5.8S were detected. We take this as indication for substantial amounts of Nog1-TAP associated particles in which the ITS2 spacer was cleaved at C2 and at least partially removed in both directions. After rpL2 depletion the 3’ trimming of 7S pre-rRNA seems to be inefficient leading to an increase in the 7S pre-rRNA to 5.8S rRNA ratio. Still, the 7S pre-rRNA signal seems to be lower in the rpL2 sample than the sum of the 5.8S rRNA and 7S pre-rRNA in the control sample. We would interpret this as an indication that the total population of Nog1 associated pre-rRNA with cleaved ITS2 is lower after rpL2 depletion than in control samples That would be reflected by the mentioned lower 25.5 + 25S rRNA in the rpL2 sample when compared to the control sample. A possible additional imbalance of (pre-)5.8S rRNA and (pre-) p25S rRNA after depletion of rpL2 might indicate that the latter is the initial target of degradation pathways (cuts, trimming ,..) in this situation. This would not be surprising considering the binding site of rpL2 in ribosomes and its observed effects on the pre-rRNA folding pathway. This could be of possible interest in future experiments and we thank reviewer #3 for sharing his observation. 

- Figures 2 and 3 needs extra labels. Please consider to indicate the location of the 5S rRNA, rpL2, rpL25, rpL34, and Nog1. Also, consider to mark the most important differences between structures.

As stated above, we now added according to reviewer’s #3 suggestions labels for 5S rRNA, Nog1, the rpL2/rpL43 heterodimer and rpL25 in Figs 2-3. Due to rpL34 being largely embedded in LSU rRNA domain III (see S1 Movie) this one is hard to label in these figures. As described above, we also introduced Movies S2-5 in the revised version of the manuscript which we believe to be helpful for comparing structures obtained from samples of control cells, including their transitions, with the ones from samples of the mutant strains.

---

## [Decision Letter · Decision Letter 1]

6 Oct 2021

PONE-D-21-15939R1Analysis of subunit folding contribution of three yeast large ribosomal subunit proteins required for stabilisation and processing of intermediate nuclear rRNA precursors.PLOS ONE

Dear Dr. Milkereit,

Thank you for submitting your manuscript to PLOS ONE. After careful consideration, we feel that it has merit but does not fully meet PLOS ONE’s publication criteria as it currently stands. Therefore, we invite you to submit a revised version of the manuscript that addresses the points raised during the review process.

Please address the residual comments by one of the reviewers.

We look forward to receiving your revised manuscript.

Kind regards,

Thomas Preiss, PhD

Academic Editor

PLOS ONE

Journal Requirements:

Additional Editor Comments (if provided):

Reviewers' comments:

Reviewer's Responses to Questions

**Comments to the Author**

1. If the authors have adequately addressed your comments raised in a previous round of review and you feel that this manuscript is now acceptable for publication, you may indicate that here to bypass the “Comments to the Author” section, enter your conflict of interest statement in the “Confidential to Editor” section, and submit your "Accept" recommendation.

Reviewer #1: (No Response)

Reviewer #3: All comments have been addressed

2. Is the manuscript technically sound, and do the data support the conclusions?

Reviewer #1: Yes

Reviewer #3: Yes

3. Has the statistical analysis been performed appropriately and rigorously? 

Reviewer #1: Yes

Reviewer #3: Yes

4. Have the authors made all data underlying the findings in their manuscript fully available?

Reviewer #1: Yes

Reviewer #3: Yes

5. Is the manuscript presented in an intelligible fashion and written in standard English?

Reviewer #1: Yes

Reviewer #3: Yes

6. Review Comments to the Author

Reviewer #1: The added labels as well as the movies have improved the manuscript. Still, many parts of the text remain difficult to follow as the Figures are still not sufficiently labelled.

Here are some examples:

"In state Nog1TAP-E (see Fig 2B, Fig 3B and Figure F in S1 Appendix) a group of four factors (Rpf1, Nsa1, Rrp1, Mak16) was possibly released from the subunit. In contrast to state Nog1TAP-F, these could not anymore be detected bound to the rRNA expansion segment ES7 and LSU rRNA domains I and II at the subunit solvent surface."

I tried to identify these factors in the respective regions, However, I was not able to find these proteins supposed to be missing in State E compared to state F. It would be helpful to have e.g.arrows in State E pointing at the positions where proteins have left, or alternatively arrows in State F indicating the proteins that are leaving during the transition from F to E.

"Otherwise, a hallmark of state Nog1TAP-E was the appearance of density unambiguously attributable to LSU rRNA domain III and many of the r-proteins associated with it. Among them were rpL25 and rpL34,…"

While rpL25 is labelled in Nog1-TAP-E particles, rpL34 is not (it is only labelled in later particles), therefore it is difficult to follow the descriptions in the text.

"Several factors (Ytm1, Erb1, Noc3, Ebp2, Brx1, Spb1, Nop16) previously associated with LSU rRNA domains I, II and III were not anymore observed and thus possibly released or with flexible orientation. Various other factors (Rrs1, Rpf2, Rsa4, Nog2, Nop53, Cgr1) and r-proteins (rpL2, rpL43, rpL5, rpL11, rpL21) newly appeared in Nog1TAP-C in association with the emerging rRNA domains."

Same as before – without labeling, it is difficult to follow this text.

I'm aware that the main point of this paper is not the wild-type particles (where much is already known) but changes observed in the particles after ribosomal protein depletion. Still, I believe it is important to first understand the maturation steps of the wild-type particles (which is only possible with better labeling), in order to be able to better compare to the alterations in ribosomal protein-depleted particles.

Reviewer #3: (No Response)

7. PLOS authors have the option to publish the peer review history of their article (what does this mean?). If published, this will include your full peer review and any attached files.

Reviewer #1: No

Reviewer #3: No

---

## [Author Response · Author response to Decision Letter 1]

12 Oct 2021

Dear Dr. Preiss, dear reviewers,

We thank you again for taking the time to inspect the manuscript and for the constructive suggestions to improve its quality.

Reviewer 1 raises the point that the labelling of the figures could still be improved (see below). We agree with reviewer 1 that parts of the text describing changes in the protein composition between different pre-ribosomal populations cannot be easily followed by using the visualization of the structure models shown in Figures 2-6. Due to the large number (twelve) of the described complex models these provide primarily an overview on the general observed rRNA fold from two defined orientations. It is often very challenging to point here on one specific protein out of the easily more than 40 protein components, with many of them embedded into the RNA and intertwined with each other. Besides providing the readers with all pdb models and density maps (currently all with status “hold for publication” in www.pdb.org), the 2D-Maps in S1 Appendix were thought to enable an easier access to the 3D-structure data by showing for each of the twelve models the protein composition, the modeled RNA helices and a simplified 2D-interaction network between them. To follow, as suggested by reviewer 1, the changes in protein compositions also in 3D-representations of the structure models we created for the new revised version an additional supplementary Figure (S9 Appendix). Here, the structure models are visualized in 14 snapshots each with optimized orientation, coloring and zoom level to highlight specific observations raised in the text. We changed the manuscript accordingly, in which we refer now to this additional Figure at several points. We refer now also at a few more points to S1 Appendix with its 2D summary of the structure models. In addition, we corrected a wrong reference on page 5 (line 254) to 5S rRNA helices 5 and 6, now referring to 5S rRNA helices 4 and 5. 

We think that, thanks to the reviewer’s suggestions, the quality of the manuscript could be thereby again improved and hope that it is now acceptable for publication in Plos One.

Reviewer 1 suggestions: 

“The added labels as well as the movies have improved the manuscript. Still, many parts of the text remain difficult to follow as the Figures are still not sufficiently labelled.

Here are some examples:

"In state Nog1TAP-E (see Fig 2B, Fig 3B and Figure F in S1 Appendix) a group of four factors (Rpf1, Nsa1, Rrp1, Mak16) was possibly released from the subunit. In contrast to state Nog1TAP-F, these could not anymore be detected bound to the rRNA expansion segment ES7 and LSU rRNA domains I and II at the subunit solvent surface."

I tried to identify these factors in the respective regions, However, I was not able to find these proteins supposed to be missing in State E compared to state F. It would be helpful to have e.g.arrows in State E pointing at the positions where proteins have left, or alternatively arrows in State F indicating the proteins that are leaving during the transition from F to E.

"Otherwise, a hallmark of state Nog1TAP-E was the appearance of density unambiguously attributable to LSU rRNA domain III and many of the r-proteins associated with it. Among them were rpL25 and rpL34,…"

While rpL25 is labelled in Nog1-TAP-E particles, rpL34 is not (it is only labelled in later particles), therefore it is difficult to follow the descriptions in the text.

"Several factors (Ytm1, Erb1, Noc3, Ebp2, Brx1, Spb1, Nop16) previously associated with LSU rRNA domains I, II and III were not anymore observed and thus possibly released or with flexible orientation. Various other factors (Rrs1, Rpf2, Rsa4, Nog2, Nop53, Cgr1) and r-proteins (rpL2, rpL43, rpL5, rpL11, rpL21) newly appeared in Nog1TAP-C in association with the emerging rRNA domains."

Same as before – without labeling, it is difficult to follow this text.

I'm aware that the main point of this paper is not the wild-type particles (where much is already known) but changes observed in the particles after ribosomal protein depletion. Still, I believe it is important to first understand the maturation steps of the wild-type particles (which is only possible with better labeling), in order to be able to better compare to the alterations in ribosomal protein-depleted particles.”

We thank reviewer 1 for his comments on the revised version of the manuscript. As described above, to provide a clean way to follow the changes in protein compositions in wildtype cells by 3D structure model representations we added now supplementary Appendix 9 to the manuscript. Here, 14 snapshots of the structure models are shown with optimized orientation, zoom level and specific coloring of protein groups of interest to visualize all the above mentioned and several other observations described in the text.

---

## [Editor Report · Decision Letter 2]

18 Oct 2021

Analysis of subunit folding contribution of three yeast large ribosomal subunit proteins required for stabilisation and processing of intermediate nuclear rRNA precursors.

PONE-D-21-15939R2

Dear Dr. Milkereit,

We’re pleased to inform you that your manuscript has been judged scientifically suitable for publication and will be formally accepted for publication once it meets all outstanding technical requirements.

Kind regards,

Thomas Preiss, PhD

Academic Editor

PLOS ONE
---

## [Editor Report · Acceptance letter]

15 Nov 2021

PONE-D-21-15939R2 

Analysis of subunit folding contribution of three yeast large ribosomal subunit proteins required for stabilisation and processing of intermediate nuclear rRNA precursors. 

Dear Dr. Milkereit:

I'm pleased to inform you that your manuscript has been deemed suitable for publication in PLOS ONE. Congratulations! Your manuscript is now with our production department. 

Kind regards, 

on behalf of

Prof Thomas Preiss 

Academic Editor

PLOS ONE